# Improving lake mixing process simulations in the Community Land Model by using K profile parameterization

Qunhui Zhang[1], Jiming Jin[1,2], Xiaochun Wang[3], Phaedra Budy[4,2], Nick Barrett[2], Sarah E. Null[2]

[1]College of Water Resources and Architectural Engineering, Northwest A & F University, Yangling, Shaanxi 712100, China

[2]Department of Watershed Sciences, Utah State University, Logan, Utah 84322, USA

[3]JIFRESSE, University of California, Los Angeles, 90095, USA

[4]US Geological Survey, Utah Cooperative Fish & Wildlife Research Unit, Logan, Utah 84322, USA

*Correspondence to*: Jiming Jin (jiming.jin@usu.edu)

**Abstract.** We improved lake mixing process simulations by applying a vertical mixing scheme, K profile parameterization

(KPP), in the Community Land Model (CLM) version 4.5, developed by the National Center for Atmospheric Research. Vertical mixing of the lake water column can significantly affect heat transfer and vertical temperature profiles. However, the current vertical mixing scheme in CLM requires an arbitrarily enlarged eddy diffusivity to enhance water mixing. The coupled CLM-KPP considers a boundary layer for eddy development and in the lake interior, water mixing is associated with internal wave activity and shear instability. We chose a lake in Arctic Alaska and a lake on the Tibetan Plateau to evaluate

this improved lake model. Results demonstrated that CLM-KPP reproduced the observed lake mixing and significantly improved lake temperature simulations when compared to the original CLM. Our newly improved model better represents the transition between stratification and turnover. This improved lake model has great potential for reliable physical lake process predictions and better ecosystem services.

## 1 Introduction

Lake thermal processes are vital to improving our understanding of regional climate systems. Lakes significantly affect regional temperature, precipitation, and surface heat fluxes (Jeffries et al., 1999; Lofgren, 2004; Long et al., 2007; Rouse et al., 2008; Thiery et al., 2015). In fact, lakes can reduce diurnal temperature variation by cooling near-surface air temperature during the day and warming it at night (Bonan, 1995; Krinner, 2003; Samuelsson et al., 2010). Regional climate modeling has shown that lakes can have a strong effect on seasonal precipitation (Diallo et al., 2017; Zhu et al., 2017). For instance,

lakes cool the lower atmosphere during the summer and increase its stability, reducing summer precipitation as compared to the land (Gu et al., 2016; Sun et al., 2015). Additionally, large lakes, like the Great Lakes in North America, often produce strong snowstorms during early winter or spring from high surface evaporation (Dai et al., 2018; Laird et al., 2009). Furthermore, Rouse et al. (2005) indicated that lakes affect surface energy balance, with higher net radiation, subsurface heat storage, and evaporation than the nearby land.

Lake temperatures shape lake ecosystems (Marshall et al., 2013; Michalski and Lemmin, 1995). For example, Berger et al. (2006) showed that plankton biomass is negatively correlated with lake mixed layer depth. Some studies have proven that strong temperature stratification stimulates the spring phytoplankton bloom (Chiswell, 2011; Mahadevan et al., 2012). What is more, the frequency and intensity of water turnover, a product of the thermal processes within a lake, is critical for replenishing and circulating hypolimnetic $O_2$ and nutrients (Dodson, 2004; Foley et al., 2012; Shimoda et al., 2011). Stratification plays an important role in lake production and food webs. Stratification and warmer epilimnion temperatures create conditions necessary for phytoplankton production. Also, when Arctic lakes become strongly stratified, the hypolimnion can become anoxic, which in turn increases nutrient recycling and leads to elevated production the following spring (O'Brien et al., 2005). Increased food availability and warmer lake temperatures in the epilimnion from stratification increase arctic char growth. Finally, simulations of stratification date and epilimnion temperature are used in bioenergetic models to estimate fish growth and consumption and better understand Arctic char production with global environmental change (Budy and Luecke, 2014). Hence, it is important to accurately quantify lake thermal processes in order to fully comprehend how temperatures affect lake ecosystems.

Numerical models are important tools for investigating lake thermal processes. Vertical mixing processes need to be parameterized in these models. The usefulness of these models depends on whether they can represent lake processes accurately and in a dynamic consistent manner. Several one-dimensional (1-D) lake models have been developed over the last three decades with varying levels of sophistication in terms of how model physics and structure are represented (Henderson-Sellers, 1985; Hostetler and Bartlein, 1990; Goudsmit et al., 2002; Mironov, 2008; Stepanenko et al., 2016). The Lake Model Inter-comparison Project (LakeMIP) assessed the simulation skill of different models (Stepanenko et al., 2010) and concluded that no single lake model is capable of simulating thermal processes for a wide range of lakes with different depths (Kheyrollah Pour et al., 2012; Stepanenko et al., 2014; Martynov et al., 2010; Perroud et al., 2009; Yao et al., 2014). Stepanenko et al. (2012) indicated that the poor skill in modeling lake thermal processes was due to the simplification of water mixing processes. Perroud et al. (2009) showed that insufficient water mixing weakened heat transfer within the lake, resulting in unrealistic temperature profile simulations. Hence, efforts have been made to improve lake mixing simulations through enlarged eddy diffusivity (Gu et al., 2013; Perroud et al., 2009). However, such an approach mostly strengthens mixing in the entire water body, which often greatly overestimates water mixing in the lower part of lakes (Subin et al., 2012; Zhang et al., 2018).

K profile parameterization (KPP) (Large et al., 1994), an advanced water mixing scheme used mostly in ocean models, significantly improves oceanic water mixing simulations (Li et al., 2001; Roekel et al., 2018; Shchepetkin and McWilliams, 2005; Wang et al., 2013). In KPP, eddy diffusivity is estimated separately for the lake boundary layer and lake interior. It considers a boundary layer for eddy development, and explicit inclusion of arbitrarily enlarged eddy diffusivity is avoided. The objective of this study is to improve lake mixing process simulations by using KPP with the Community Land Model

(CLM) version 4.5, developed by the National Center for Atmospheric Research (Oleson et al., 2013). This newly improved model was then applied to an Arctic Alaskan lake and a lake called Nam Co in the Tibetan Plateau (TP) for model verification. In this paper, Sect. 2 introduces the mixing schemes, data, and methodology, Sect. 3 presents simulation results and analysis, and conclusions and discussion are given in Sect. 4.

## 2 Mixing schemes, data, and methodology

### 2.1 Mixing scheme descriptions

#### 2.1.1 The original mixing scheme in the CLM lake model

The 1-D lake model embedded in the current CLM version (CLM-ORG) simulates heat and water exchanges between the air and lake surface, water phase changes, and radiation transfer and water mixing within the lake. The lake model consists of up to 5 snow layers on the lake ice, 10 water and ice layers, 10 soil layers, and 5 bedrock layers. Mixing processes in CLM-ORG contain wind-driven eddy diffusion, an enhanced diffusion, molecular diffusion, and convective mixing. The convective adjustment scheme is activated when there is an unstably stratified water column (Hostetler and Bartlein, 1990). The first three diffusion terms are included in the water diffusivity parameterization. The total diffusivity in the lake model is calculated as follows (Subin et al., 2012):

$$K_w^{ORG} = m_d(\kappa_e + K_{ed} + \kappa_m) \tag{1}$$

where $\kappa_e$ represents wind-driven diffusivity (m$^2$ s$^{-1}$), $K_{ed}$ is the enhanced eddy diffusivity to strengthen mixing processes (m$^2$ s$^{-1}$), $\kappa_m$ is a constant molecular diffusivity equal to $1.4 \times 10^{-7}$ m$^2$ s$^{-1}$, and $m_d$ is a parameter to increase the diffusivity for deep lakes, which is equal to 10 when lake depth is greater than 25 m. Wind-driven diffusivity, $\kappa_e$, is formulated as follows:

$$\kappa_e = \begin{cases} \dfrac{\kappa w^* z}{P_0(1 + 37R_i^2)} \exp(-k^* z), & T_g > T_f \\ 0, & T_g \leq T_f \end{cases} \tag{2}$$

where $T_g$ is the water surface temperature (WST) (K); $T_f$ is the freezing temperature, equal to 273.15 K; $\kappa$ is the von Karman constant; $P_0$ is the turbulent Prandtl number, equal to 1; and $z$ is depth, which increases downward (m). $w^*$ is the surface friction velocity (m s$^{-1}$) calculated as:

$$w^* = C_d u_2 \tag{3}$$

where $u_2$ is the 2 m wind speed (m s$^{-1}$), and $C_d$ is the drag coefficient equal to 0.0012. $k^*$ is related to latitude $\varphi$:

$$k^* = 6.6 u_2^{-1.84} \sqrt{|\sin\varphi|} \tag{4}$$

$R_i$ is the Richardson number, given as:

$$R_i = \frac{-1 + \sqrt{1 + \dfrac{40N^2 \kappa^2 z^2}{w^{*2}\exp(-2k^* z)}}}{20} \tag{5}$$

where $N$ is the local buoyancy frequency representing the stability of water (s[-1]),

$$N^2 = \frac{g}{\rho}\frac{\partial \rho}{\partial z} \tag{6}$$

$g$ is gravity acceleration (m s[-2]), and $\rho$ is the density of water (kg m[-3]). The equation of the enhanced diffusivity is:

$$K_{ed} = 1.04 \times 10^{-8}(N^2)^{-0.43}, (N^2 \geq 7.5 \times 10^{-5}\ s^{-2}) \tag{7}$$

When $N^2$ reached at least 7.5×10[-5] s[-2], the enhanced diffusivity is about six times greater than the molecular diffusivity (Fang and Stefan, 1996). The wind-driven diffusivity is typically at least 2 orders larger than the molecular diffusivity (Hostetler and Bartlein, 1990). Thus, winds have a dominant effect on water mixing in the CLM lake model. In practical application, the total diffusivity computed by Eq. (1) generally produces unrealistically weak mixing and causes large errors in temperature profile simulations (Gu et al., 2013; Zhang et al., 2018).

### 2.1.2 KPP

KPP has two different diffusivity parameterizations for the lake boundary layer and the layer below, which is different from the total diffusivity represented in the original CLM lake model. The diffusivity of the lake boundary layer, a function of surface forcing and the lake boundary layer depth, is based on the Monin-Obukhov similarity theory (Monin and Obukhov, 1954):

$$K_w^{KPP}(\sigma) = hw(\sigma)G(\sigma) + \kappa_m \tag{8}$$

where $\sigma = d/h$ is the dimensionless vertical coordinate varying from 0 at the lake surface to 1 at the bottom of the lake boundary layer $h, w(\sigma)$ is the velocity scale, and $G(\sigma)$ is the shape function. $\kappa_m$ is a constant molecular diffusivity (m[2] s[-1]), as in Eq. (1). The velocity scale is:

$$w(\sigma) = \begin{cases} \dfrac{\kappa u^*}{\varnothing\left(\dfrac{\varepsilon h}{L}\right)}, \varepsilon < \sigma < 1, \zeta < 0 \\[12pt] \dfrac{\kappa u^*}{\varnothing(\dfrac{\sigma h}{L})}, \qquad otherwise \end{cases} \tag{9}$$

where $\kappa$ is the von Karman constant (0.4), $\varepsilon$ is equal to 0.1, and $u^*$ is the surface friction velocity (m s[-1]) calculated as (Large and Pond, 1982):

$$u^{*2} = \frac{\rho_a}{\rho}C_d U^2 \tag{10}$$

$$10^3 C_d = \frac{2.70}{U} + 0.142 + 0.0764U \tag{11}$$

where $\rho_a$ and $\rho$ are the air and lake water densities (kg m[-3]), respectively, $C_d$ is the drag coefficient, and $U$ is the 10 m wind speed (m s[-1]). $\varnothing(\zeta)$ is a non-dimensional flux profile associated with the stability parameter $\zeta = d/L = \sigma h/L$, and $L$ is the Monin-Obukhov length scale defined as:

$$L = u^{*3}/\kappa B_f \tag{12}$$

where $B_f$ is the buoyancy flux (m$^2$ s$^{-3}$):

$$B_f = H^* g \alpha C_p^{-1} \rho^{-1} \tag{13}$$

$H^*$ is the sum of the surface turbulent heat fluxes, net long-wave radiation, and net shortwave radiation for the lake boundary layer (W m$^{-2}$), $\alpha$ is the constant thermal expansion coefficient, and $C_p$ is the specific heat capacity of water (J kg$^{-1}$ K$^{-1}$). The non-dimensional shape function $G(\sigma)$ is a third-order polynomial (see the Appendix).

Water mixing below the lake boundary layer considers vertical shear and internal waves. The equation is:

$$K_w^{KPP} = k_s + k_w + \kappa_m \tag{14}$$

where $k_s$ is the diffusivity due to shear instability (m$^2$ s$^{-1}$), and $k_w$ is the internal wave diffusivity set to a constant ($10^{-7}$ m$^2$ s$^{-1}$) as the background diffusivity (Bryson and Ragotzkie, 1960; Powell and Jassby, 1974; Thorpe and Jiang, 1998). The shear mixing term is calculated as:

$$k_s = \begin{cases} k_0, & Ri_g < 0 \\ k_0[1 - (Ri_g/Ri_0)^2]^p, & 0 < Ri_g < Ri_0 \\ 0, & Ri_0 < Ri_g \end{cases} \tag{15}$$

where $k_0 = 10^{-5}$ m$^2$ s$^{-1}$ (Etemad-Shahidi and Imberger, 2006; Sweers, 1970), $Ri_0 = 0.7$, and $p = 3$. $Ri_g$ is the local gradient

Richardson number:

$$Ri_g = \frac{N^2}{\left(\frac{\partial V}{\partial z}\right)^2} \tag{16}$$

$$V = V_{sfc}\left(3\left(\frac{z}{D}\right)^2 - 4\left(\frac{z}{D}\right) + 1\right) \tag{17}$$

$$V_{sfc} = 0.028W \tag{18}$$

where $V$ is the horizontal velocity of water (m s$^{-1}$), $D$ is the lake depth (m), $V_{sfc}$ is the surface water flow velocity (m s$^{-1}$), and $W$ is the surface wind (m s$^{-1}$). To apply KPP in the CLM lake model, we use Eq. (17) to represent the change of water flow in the vertical direction over the entire lake depth $D$ (Banks, 1975; Verhagen, 1994). We can see in Eq. (18) that $V_{sfc}$ is linked with $W$ (Stanichny et al., 2016; Wu, 1975).

The boundary layer depth depends mainly on the buoyancy and horizontal water flow velocity profiles. In order to compute the boundary layer depth, the bulk Richardson number is first computed as follows:

$$Ri_b(d) = \frac{(B_r - B(d))d}{|V_r - V(d)|^2 + V_t^2(d)} \tag{19}$$

where $Ri_b$ is the bulk Richardson number, and $B$ is the buoyancy. When $Ri_b$ is equal to 0.25 (Kunze et al., 1990; Peters et al., 1995), the shallowest water depth $(d)$ is treated as the depth of the lake boundary layer. The subscript $r$ represents the near-surface water layer with a depth of 0.1 m ($B_r, B(d), V_t^2(d)$, see the Appendix).

In this study, KPP was implemented into the CLM lake model (CLM-KPP) to improve lake mixing process simulations. In KPP, eddy diffusivity is estimated separately for the lake boundary layer and lake interior. In the lake boundary layer, the eddy diffusivity is determined not by the local gradient of mean variables, but by surface forcing and the boundary layer

depth. The non-local effect is taken into account by estimating the boundary layer depth first, and eddy diffusivity is then specified with a prescribed profile in the lake boundary layer. In the lake interior, mixing is associated with internal wave activity and shear instability. However, CLM-ORG does not consider a boundary layer for eddy development, and insufficient water mixing is enhanced through an ad hoc parameter, which is often unable to reflect reality (Zhang et al., 2018). Thus, the coupling of CLM-KPP is essential for better understanding of lake mixing processes.

## 2.2 Study area

We selected two lakes with available data to evaluate the original lake mixing scheme and KPP. Fog3 Lake is in Arctic Alaska at (68.67° N, 149.10° W) (Fig. 1a). In 2018 it had a surface area of 38,863 m$^2$ and a maximum depth of 21 m. The lake has a long ice duration, and ice-off is usually in late June, while ice-on typically occurs in early October (Arp et al., 2015). Around this lake, the mean annual air temperature is about ~ –6 ℃, and the mean annual precipitation is ~ 200 mm (Ping et al., 1998). This kettle lake is surrounded by lower hills covered mainly with shrubs and tundra. Due to the treeless landscape, there are no shielding effects on the wind. In addition, Fog3 Lake is formed by glaciers, and has less connection to other surrounding surface waters. The second lake is Nam Co, the highest and largest lake in the central TP (Fig. 1b). It is situated over 30.5–30.95° N, 90.2–91.05° E with an altitude of 4,730 m and a surface area of about 2,021 km$^2$ in 2010 (Lei et al., 2013; Zhu et al., 2010). Its maximum depth reaches more than 95 m, and the mean depth is about 40 m (Wang et al., 2009). The main water supply to Nam Co is precipitation and melting glaciers. Nam Co is a closed lake with no outflow, and water loss occurs mainly through evaporation (Ma et al., 2016).

## 2.3 Data

Observed hourly meteorological station data for Fog3 Lake were used to drive CLM-ORG and CLM-KPP. Fog3 Lake is about 1.5 km from Toolik Field Station (68°37.796' N, 149°35.834' W), in the northern foothills of the Brooks Mountain Range, Alaska (https://toolik.alaska.edu/edc/abiotic_monitoring/index.php). The forcing variables include downward shortwave and longwave radiation, wind speed, air temperature, air pressure, and specific humidity. Observed lake temperatures from 1 July through 31 August 2018 are for lake depths of 0, 1, 2, 3, 4, 5, 6, 7, 8, 10, 12, 14, and 16 m for model initialization and evaluation.

For Nam Co, the forcing data were from the gridded China meteorological dataset developed by the hydro-meteorological research group at the Institute of Tibetan Plateau Research, Chinese Academy of Sciences (ITPCAS) (Chen et al., 2011; He and Yang, 2011). The forcing variables in this dataset are the same as those for the Alaskan lake. The ITPCAS data cover the period 1979–2015 with a spatial resolution of 0.1 degree and a time step of 3 h. We used the Nam Co meteorological station data for the period of October 2005 through December 2010 to assess the ITPCAS forcing variables. These forcing variables agreed very well with the Nam Co station data, except the wind speed showed significant biases (Figs. not shown). Linear

regression with the station wind speed was applied to correct these biases. Monthly Moderate Resolution Imaging Spectroradiometer (MODIS) surface temperature data at a spatial resolution of 0.05 degree (Savtchenko et al., 2004; Wan et al., 2010) were applied to evaluate the model results for Nam Co. Previous studies have verified MODIS WST data for lakes with *in situ* observations (Crosman and Horel, 2009; Schneider et al., 2009). Zhang et al. (2014) found that the nighttime

WST of MODIS for Nam Co had a 0.89 correlation coefficient and a −1.4 ℃ bias when compared with surface observations. All these studies show that the MODIS WST has acceptable accuracy for studying lake thermal processes.

## 2.4 Experiment design

Simulations for Fog3 Lake were conducted with both CLM-ORG and CLM-KPP from 1 July through 31 August 2018. The depth for this lake was set at 20 m in both models. Observed lake temperatures for Fog3 Lake are for lake depths of 0, 1, 2, 3,

4, 5, 6, 7, 8, 10, 12, 14, and 16 m. The lake model has 10 lake layers by default, and the center point depths of these layers are 0.05, 0.3, 0.9, 1.9, 3.3, 5.1, 7.5, 10.3, 13.79, and 17.94 m generated automatically by the layering scheme in the model based on the input lake depth. For this study, we tried to keep each layer thin in the top part of the lake to reflect diurnal cycles (layers 1–5) in both CLM-ORG and CLM-KPP. Below layer 5, we used mostly the observed points to layer the rest of the lake column. Finally, we produced 24 layers for the entire lake column in both models, and the center point depths of

these lake layers are 0.05, 0.15, 0.25, 0.35, 0.45, 1, 2, 3, 4, 5, 6, 7, 8, 9, 10, 11, 12, 13, 14, 15, 16, 17, 18, and 19.25 m, respectively. The lake temperatures were initialized with observations for 1 July 2018. The WST and temperature profile simulations with CLM-ORG and CLM-KPP were compared with the observed lake temperatures. For Nam Co, we also used 10 default layers for lake depths in our models without observed vertical temperature profiles, and lake depths were set based on observations (Wang et al., 2009), which ranged from 20 to 95 m. There were 34 model grid cells covering Nam Co with a

spatial resolution of 0.1 degree. The temperature of each layer was initialized with 277 K. The simulated period for Nam Co was from 2001 through 2012. The simulations for the first two years were discarded as model spin-up, and the remaining simulations were used for analysis. The metrics used for evaluating the performance of the model included the root mean square error (RMSE) and correlation coefficient (R).

## 3 Results

### 3.1 Simulations for Fog3 Lake with CLM-ORG and CLM-KPP

WST simulations with CLM-KPP were more accurate than those with CLM-ORG, especially in August. The RMSE of WST decreased from 0.8 ℃ with CLM-ORG to 0.4 ℃ with CLM-KPP (Fig. 2). CLM-KPP also produced better vertical lake temperature profile simulations than CLM-ORG, particularly in mid to late August. The observations showed that the lake mixed on 16 August (Fig. 3a). CLM-KPP accurately captured the mixing event (Fig. 3c), while CLM-ORG produced strong

stratification in the upper part of the lake throughout the simulation period (Fig. 3b). Insignificant differences were seen

between CLM-ORG and CLM-KPP when compared to observations for the period before 16 August (Table 1), while remarkable improvements were achieved with CLM-KPP during 16–31 August after a strong wind event occurred (Figs. 3d–e). The RMSE of the temperature profile simulations decreased from 1.4 ˚C with CLM-ORG to 0.3 ˚C with CLM-KPP, and R increased from 0.57 to 0.99 for 16–31 August 2018 (Table 1). In general, CLM-KPP had superior performance in simulating well-mixed conditions when compared with CLM-ORG, indicating a successful implementation of KPP into CLM.

Simulations of total diffusivity (m$^2$ s$^{-1}$) $K_w^{KPP}$ with CLM-KPP were compared with those of $K_w^{ORG}$ with CLM-ORG. $K_w^{KPP}$ within the boundary layer was generally larger than $K_w^{ORG}$, especially in August (Fig. 4). However, the total diffusivity with CLM-ORG was higher than that with CLM-KPP below the boundary layer (Fig. 4). The pattern of the diffusivity with CLM-ORG was consistent with that of the squared buoyancy frequency $N^2$ (Fig. 5), implying that the enhanced diffusivity ($K_{ed}$) was weighted very highly in $K_w^{ORG}$ in this model. In the meantime, $K_w^{KPP}$ was mostly on the order of $10^{-7}$ m$^2$ s$^{-1}$ and was the sum of internal-wave diffusivity, molecular diffusivity, and diffusivity due to shear instability (Eq. (14)). The first two terms were also on the order of $10^{-7}$ m$^2$ s$^{-1}$, indicating that the total diffusivity with CLM-KPP was controlled mostly by these two terms. In early July, $K_w^{KPP}$ sometimes appeared to be on the order of $10^{-5}$ m$^2$ s$^{-1}$, which was consistent with that of the last term, shear instability diffusivity, implying that this term dominated $K_w^{KPP}$. The diffusivity increase was closely related to the strong winds occurring at the same time (Fig. 4b).

The squared buoyancy frequency $N^2$ of simulations with both CLM-KPP and CLM-ORG were also compared for our study period. $N^2$ was related to the water density gradient (Eq. (6)) determined by the temperature gradient in both models. A greater $N^2$ produced more stable water and stronger water stratification. From 1 July through 15 August, the simulated $N^2$ with CLM-KPP near the bottom of the boundary layer was slightly larger than that with CLM-ORG (Fig. 5). Thus, the simulated water stratification with CLM-KPP at the bottom of the boundary layer was stronger than that in CLM-ORG before 16 August. However, after 16 August, the maximum $N^2$ with CLM-ORG occurred in the middle layer of the lake, maintaining stratification there. Conversely, the maximum $N^2$ with CLM-KPP moved down to near the bottom of the lake during the same 16-day period (Fig. 5).

### 3.2 Analysis of CLM-KPP simulations for Fog3 Lake

We examined our simulations and meteorological forcing data in detail to physically understand water mixing conditions simulated by CLM-KPP, especially over the period of 16–31 August 2018. Figure 6a shows that downward shortwave radiation was 45 W m$^{-2}$ less during 1–15 August (shaded area) than in July. Meanwhile, over the same period, air temperature and specific humidity decreased dramatically, while wind speed showed almost no trend (Figs. 6b–d). In this period, the simulated net radiation with CLM-KPP was 54 W m$^{-2}$ lower than that for July (Fig. 6e). The turbulent heat flux, the sum of sensible and latent heat fluxes, increased over this 15-day period due mainly to the decreased air temperature and

humidity (Fig. 6f). Figure 6g shows that buoyancy flux, defined as net radiation minus turbulent heat flux in the boundary layer with a different unit ($m^2 \ s^{-3}$), was mostly negative during 1–15 August, showing that the lake was losing heat. Due to this heat loss, the temperature in the upper lake decreased, reducing the temperature difference between the upper and lower parts of the lake and thus weakening the stratification. Therefore, we can see that the boundary layer depth increased over the period of 1–15 August (Fig. 6h) when the wind had no systematic changes, but the buoyancy flux played a significant role in this increase.

During 15–16 August, a wind event (12 m s$^{-1}$) mixed the lake, dramatically increasing the boundary layer depth in addition to the negative buoyancy flux. The deep boundary layer was maintained through the end of August, even though the winds returned to normal conditions. Such strong mixing was not seen in CLM-ORG, where the water stratification could not be broken up by the high wind event without help from the negative buoyancy flux. Hence, without an ad hoc parameter to enhance the water diffusivity as in CLM-ORG, CLM-KPP still reproduced the observed water mixing processes.

**3.3 Model validation with Nam Co data**

We validated both CLM-ORG and CLM-KPP with MODIS data for Nam Co by conducting 10-km spatial resolution simulations for this lake over the period of 2003 through 2012. We can see that CLM-KPP improved WST simulations averaged over the entire lake (34 model grid cells) when compared with the MODIS data and CLM-ORG simulations (Fig. 7). The RMSE of WST decreased from 4.58 ˚C with CLM-ORG to 2.23 ˚C with CLM-KPP, and R increased from 0.90 to 0.96 at the same time.

The improved WST simulations with CLM-KPP were closely related to the water diffusivity simulations with KPP as discussed above. We averaged the $K_w^{ORG}$ and $K_w^{KPP}$ simulations over water columns with depth greater than 25 m for Nam Co, as shown in Fig. 8, and the total of such columns were 28 out of 34 for this lake. Figure 8 indicates that $K_w^{KPP}$ was slightly smaller than $K_w^{ORG}$ mostly in the mixing layer of the lake over the summer. The difference likely resulted from the enlarged $K_w^{ORG}$ in CLM where this parameter was increased by a factor of 10 when the lake depth was greater than 25 m. In the deeper part of the lake, $K_w^{KPP}$ was much smaller than $K_w^{ORG}$ over the summer due much to the contribution of $K_{ed}$ to $K_w^{ORG}$. In the spring and fall, $K_w^{KPP}$ was significantly larger than $K_w^{ORG}$. During the winter when the lake froze, both CLM-KPP and CLM-ORG were set to use $K_w^{ORG}$. We can see that the most significant improvements in WST for Nam Co occurred during the ice-free seasons when KPP was activated. Overall, CLM-KPP can enhance the water diffusivity during spring and fall and maintain weak water diffusivity in the lake interior during summer when stratification is strong.

**4 Conclusions and discussion**

We improved lake mixing process simulations by applying the vertical mixing scheme KPP in CLM. The improved lake model was applied to an Arctic Alaskan lake and to Nam Co lake in the TP for model evaluation. Results for the Alaskan

lake indicate that the WST and lake temperature profile simulations using KPP are greatly improved when compared to the original vertical mixing scheme in CLM. During the transition season in August, the improvement is most obvious. This improvement is associated with negative heat flux and high wind, which can cause deepening of the boundary layer and strong mixing. However, the original vertical mixing scheme of CLM cannot capture these strong mixing events and causes a

5 positive lake temperature bias in its simulation. CLM-KPP was further validated with the observed data from Nam Co, and results showed that WST simulations were significantly improved when compared with the MODIS data and CLM-ORG simulations.

More data are needed to further verify CLM-KPP, including atmospheric forcing data over lakes and observed lake temperature profiles. It should also be noted that although CLM-KPP has improved thermal process simulations, large WST

10 biases still existed during the ice freezing period for Nam Co. Such biases most likely resulted from the oversimplified lake ice scheme in the CLM lake model. Therefore, a more realistic ice scheme in lake models is needed for better understanding of the effects of water mixing on ice formation. In general, this coupled model provides an important tool for lake hydrology and ecosystem studies.

**Appendix**

Lake temperature is calculated as follows:

$$\frac{\partial T}{\partial t} = \frac{\partial}{\partial z}\left\{K_w(z,t)\frac{\partial T}{\partial z}\right\} + \frac{1}{C_w}\frac{\partial \phi}{\partial z} \tag{A1}$$

where $T$ is lake temperature (K) at depth $z$ (m) and time $t$ (s), $\phi$ is the absorbed solar radiation flux as a heat source

term (W m$^{-2}$), $C_w$ is the volumetric heat capacity of lake water (J m$^{-3}$ K$^{-1}$), and $K_w$ is the total diffusivity (m$^2$ s$^{-1}$).

The non-dimensional flux profiles are calculated as follows:

$$\emptyset = \begin{cases} 1 + 5\zeta, & 0 \leq \zeta \\ (1 - 16\zeta)^{-1/2}, & -1.0 \leq \zeta < 0 \\ (-28.86 - 98.96\zeta)^{-1/3}, & \zeta < -1.0 \end{cases} \tag{A2}$$

The non-dimensional shape function $G(\sigma)$ is a third-order polynomial:

$$G(\sigma) = a_0 + a_1\sigma + a_2\sigma^2 + a_3\sigma^3 \tag{A3}$$

$a_0$, $a_1$, $a_2$, and $a_3$ are given as:

$$a_0 = 0 \tag{A4a}$$

$$a_1 = 1 \tag{A4b}$$

$$a_2 = -2 + 3\frac{\upsilon(h)}{hw(1)} + \frac{\partial \upsilon(h)}{w(1)} + \frac{\upsilon(h)\partial w(1)}{hw(1)^2} \tag{A4c}$$

$$a_3 = 1 - 2\frac{\upsilon(h)}{hw(1)} - \frac{\partial \upsilon(h)}{w(1)} - \frac{\upsilon(h)\partial w(1)}{hw(1)^2} \tag{A4d}$$

where $\upsilon(h)$ is the total diffusivity as a function of lake depth $(h)$, $w(1)$ is the velocity scale at the bottom of the lake

boundary layer, $\partial \upsilon(h)$ is the lake depth derivative of $\upsilon$, and $\partial w(1)$ is the lake depth derivative of $w$ at the bottom of the

lake boundary layer.

$B(d)$ is the buoyancy calculated with a depth of $d$ as:

$$B(d) = g(1 - \frac{\rho_r}{\rho(d)}) \tag{A5}$$

$V_t^2$ is calculated as:

$$V_t^2(d) = \frac{C_v dN w_s(-\beta_T C_s \varepsilon)^{-1/2}}{Ri_c \kappa^2} \tag{A6}$$

where $Ri_c = 0.25$, $C_v = 1.6$, $\beta_T = 0.2$, and $C_s = -98.96$.

**Code and data availability**

The model configuration and input data used in this study are available upon request.

**Author contribution**

Qunhui Zhang conducted the modeling, performed the analysis, and drafted the manuscript; Jiming Jin designed the study, interpreted the results, and supervised the research; Xiaochun Wang contributed the original ideas of the research; Phaedra Budy and Nick Barrett provided observational data and helped with the study design and data analysis; Sarah Null gave constructive comments on the results. All authors edited the manuscript.

## 5  Competing interests

The authors declare that they have no conflict of interest.

**Acknowledgments**

This work was supported by the National Natural Science Foundation of China [grant numbers 91637209, 41571030, and 91737306] and a grant from the US National Science Foundation [grant number 1603088]. Q.Z. was also supported by the
National Key R&D Program of China on monitoring, early warning, and prevention of major natural disasters (No. 2018YFC150703). J.J. was also supported by Utah Agricultural Experiment Station.

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

**Figures**

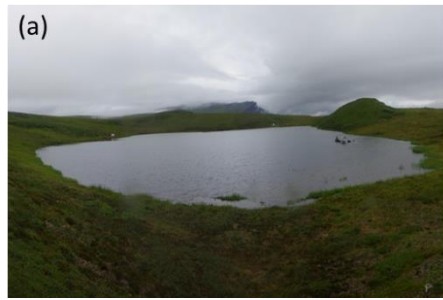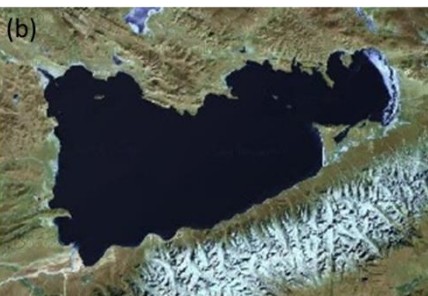

**Figure 1. (a) Fog 3 Lake and (b) Nam Co.**

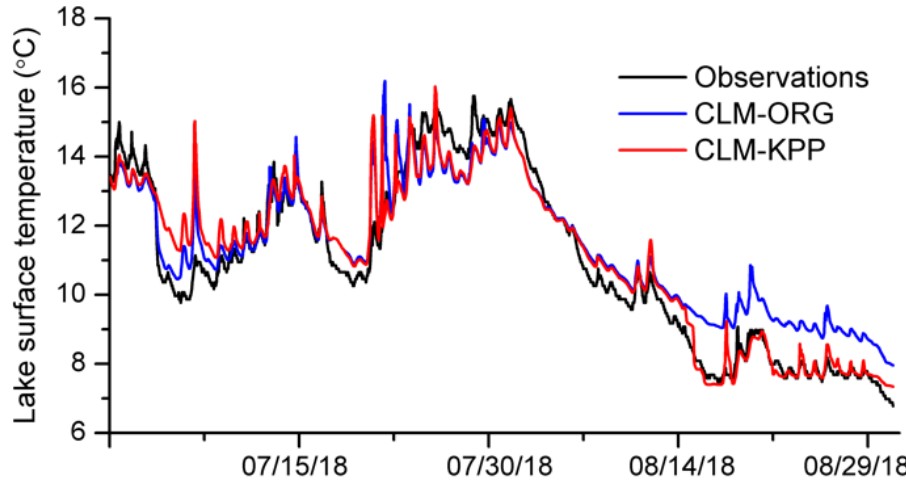

**Figure 2. WST observations (black line) and simulations with CLM-ORG (blue line) and CLM-KPP (red line) (unit: ˚C).**

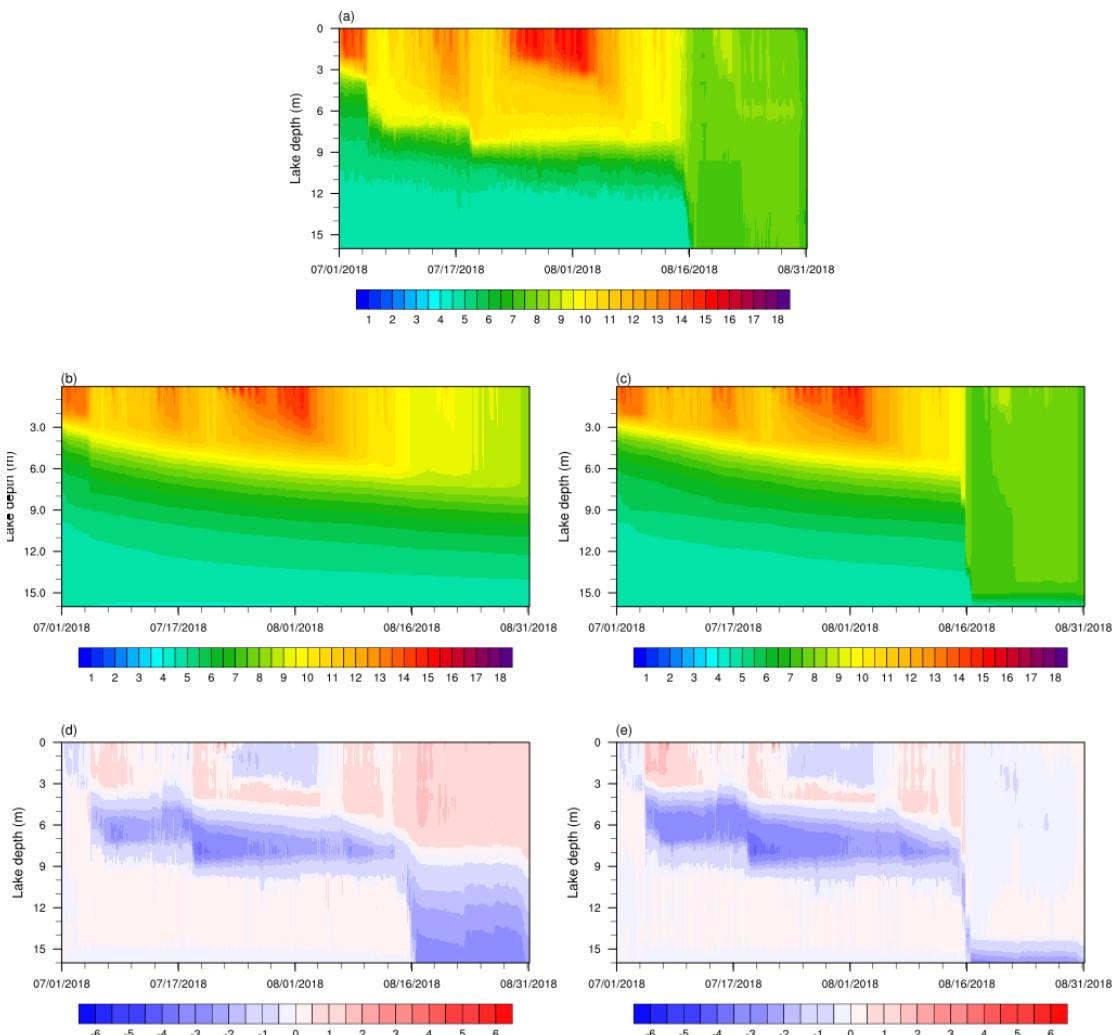

**Figure 3. Lake temperature profiles of (a) observations and simulations with (b) CLM-ORG and (c) CLM-KPP. Lake temperature profile differences between simulations and observations (d) CLM-ORG minus observations and (e) CLM-KPP minus observations (unit: ˚C).**

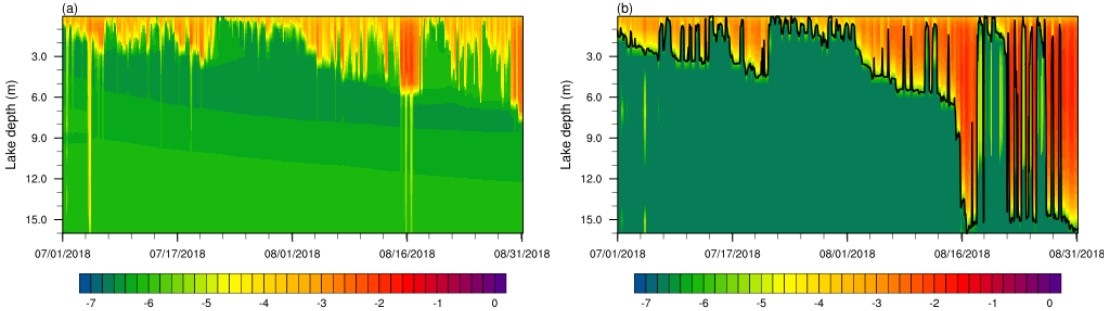

**Figure 4.** Simulated (a) $log_{10} K_w^{ORG}$ with CLM-ORG, (b) $log_{10} K_w^{KPP}$ with CLM-KPP (Unit: $m^2 s^{-1}$). The black line in (b) shows the lake boundary layer depth (Unit: m).

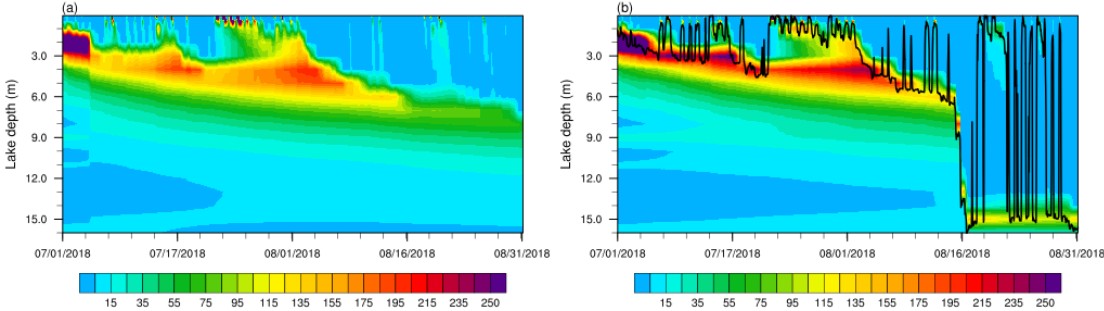

**Figure 5.** Simulated $N^2$ with (a) CLM-ORG and (b) CLM-KPP (Unit: $10^{-5}$ s$^{-2}$). The black line in (b) shows the lake boundary layer depth (Unit: m).

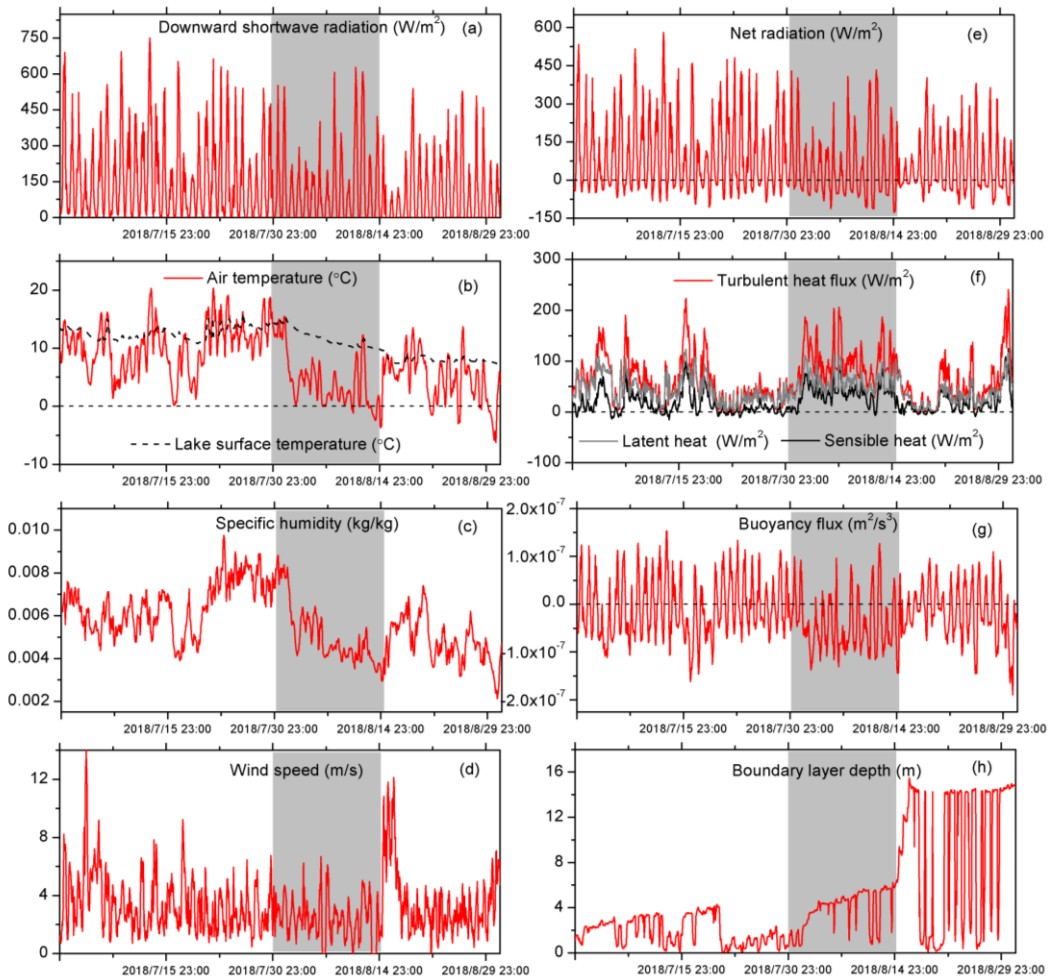

**Figure 6.** Time series of (a) observed downward shortwave radiation (W m$^{-2}$), (b) observed air temperature and WST (˚C), (c) observed specific humidity (kg kg$^{-1}$), (d) observed wind speed (m s$^{-1}$), (e) simulated net radiation (W m$^{-2}$), (f) simulated turbulent heat flux (W m$^{-2}$) (red line) with latent heat flux (gray line) and sensible heat flux (black line), (g) simulated buoyancy flux (m$^2$ s$^{-3}$), and (h) simulated boundary layer depth (m). The gray shading covers 1 August through 15 August. The simulations were from CLM-KPP.

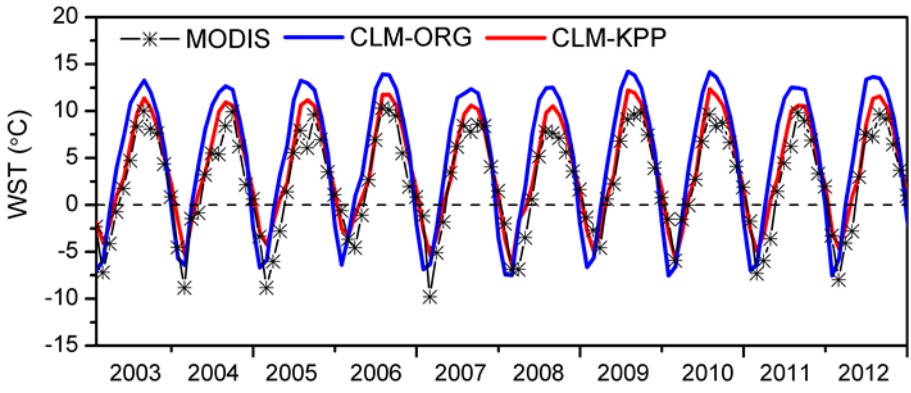

**Figure 7. Time series over the period of 2003 through 2012 of monthly WST observations from MODIS (black starred line) and simulations with CLM-ORG (blue line) and CLM-KPP (red line) (Unit: ˚C).**

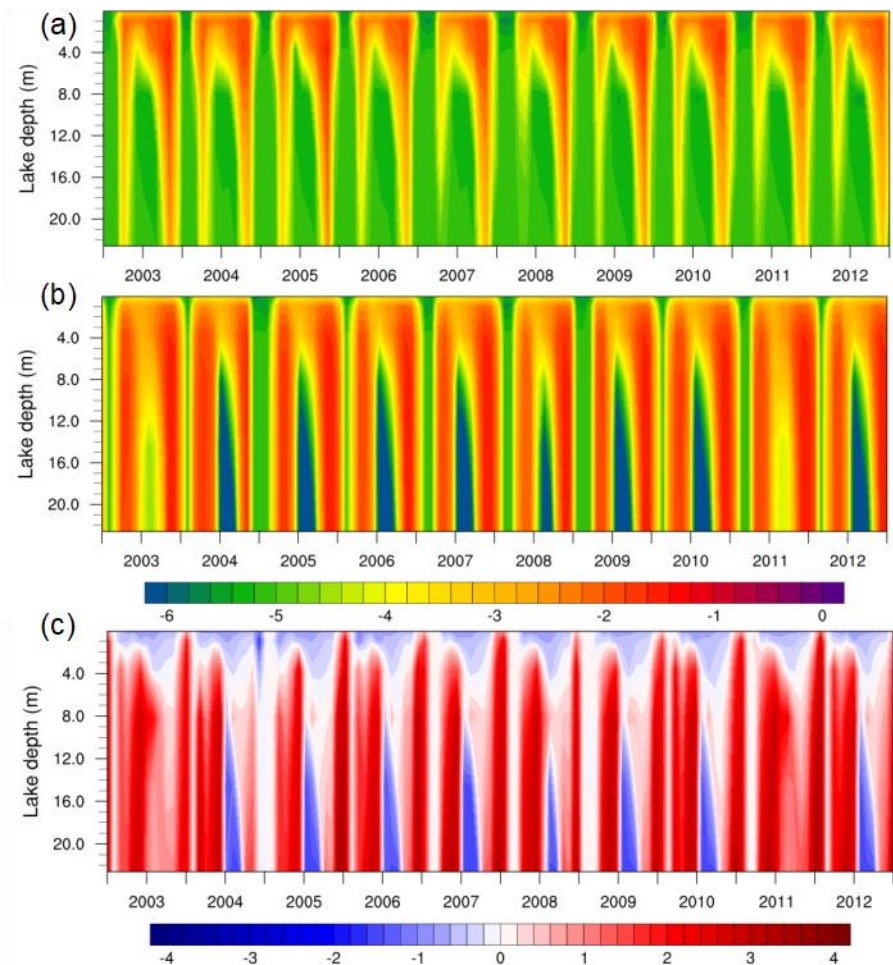

**Figure 8.** Simulated (a) $log_{10} K_w^{ORG}$ with CLM-ORG, (b) $log_{10} K_w^{KPP}$ with CLM-KPP (Unit: $m^2$ $s^{-1}$) averaged over water columns with depth greater than 25 m (28 of 34 grid cells), and (c) differences between $log_{10} K_w^{KPP}$ and $log_{10} K_w^{ORG}$ ($log_{10} K_w^{KPP} - log_{10} K_w^{ORG}$).

**Table**

**Table 1. RMSE (˚C) and R of temperature profile simulations with CLM-ORG and CLM-KPP for Fog3 Lake for the periods of 1 July–15 August and 16–31 August 2018.**

| | 1 July–15 August, 2018 | | 16–31 August, 2018 | |
|---|---|---|---|---|
| | RMSE (˚C) | R | RMSE (˚C) | R |
| CLM-ORG | 1.1 | 0.93 | 1.4 | 0.57 |
| CLM-KPP | 1.3 | 0.92 | 0.3 | 0.99 |