# Peer review of "Improving lake mixing process simulations in the Community Land Model by using K profile parameterization"

_Hydrology and Earth System Sciences, 2019_

## Short Comment (SC1) · 5 Jul 2019

It is an interesting study. Because a 1-D lake model is still much needed to understand the impact of climate changes on global lake systems, a parameterization method that could improve the simulation of lake mixing process will be much valued. But I suggest that the manuscript can be improved in the following directions.

First of all, the comparison between CLM-ORG and CLM-KPP is not exhausted, to day the least. In Subin's CLM-ORG paper, he actually tested the model over a pair of lakes around the globe. In fact, the CLM-ORG performance on high-latitude lakes which this study focused on was not the worst. Thus, the method can become much more
Interactive
comment

valuable if the authors can apply this method to some more lakes, especially those deep and large lakes.

Second, more information about the study lake is needed. Is Fog 3 Lake a glacial lake or a thermokarst lake? How was the surface friction velocity derived for this lake? Are the effects of lake fetch and wind shielding considered? What is the lake's light attenuation coefficient?

Third, how are CLM-ORG and CLM-KPP calibrated in this study? I know that CLM-ORG has a water mixing parameter that can be used to increase diffusivity for those deep lakes. Can the parameter values of CLM-KPP described here be applied to other lakes?

Forth, I am surprised that the case study did not cover the period of spring water mixing which can have large biogeochemical impacts for high-latitude lakes.

Zeli Tan (zeli.tan@pnnl.gov) Pacific Northwest National Laboratory

―――――――――――――――――――

---

## Referee Comment (RC1) · Anonymous Referee #1 · 26 Aug 2019

The manuscript presents results of introducing "K profile" parameterizatioin of turbulence into lake module of Community Land Model. This is likely the first time K profile parameterization is used in a 1D lake model, though it is widely applied in ocean models. Incorporation of new turbulence closure instead of standard Henderson-Seller diffusivity lead to significant improvement of simulation of late-summer destratification event in an Alaskan lake.

**General comment**

My general comment on the manuscript is that since single mixing event is simulated, more physical analysis could be provided to explain *why* K profile closure performed

better than Henderson-Sellers in this case. Analysis presented in sections 3.1 and 3.2 is superficial and does not touch this question. One mixing case is not enough to state that K profile is better in similar situations in general, so more substantial inquiry into physics behind both parameterizations is needed. The authors state that KPP includes effects of thermal forcing, whereas original scheme of CLM model does not. This is actually not correct. First, original CLM model includes convective adjustment scheme (Subin et al., 2012) which instantaneously mixes the unstably stratified water column. Then , the effects of stable stratification are included via Brunt-Vaisala frequency in Henderson-Sellers (H-S) diffusivity. Thus, thermal (density) stratification is taken into account. The mixing event the authors focus on happens during weakly stable stratification under strong wind forcing. One may conclude from simulation results presented is that given the same stable temperature profile the larger wind speed is needed for H-S to mix completely the water column than for KPP model. This may be elaborated by conducting idealized simulations with both turbulence closures with varying wind speeds and temperature profiles where this statement may be checked and respective quantitative estimates provided.

**Specific comments**

1) Lines 88-90: "Researchers have attempted to advance this lake model to more closely reflect reality over the last two decades (Fang and Stefan, 1996; Henderson-Sellers, 1985; Hostetler and Bartlein, 1990; Subin et al., 2012)." Three of four papers cited here do not deal with CLM model.

2) Lines 92-93: "... is the enhanced eddy diffusivity for unresolved mixing processes". All mixing processes in 1D model are unresolved and are parameterized, because only 3D model of sufficiently high resolution simulates turbulence explicitly.

3) Line 98: "0.0012 $u_2$" I guess, you can write drag coefficient $C_d$ instead of 0.0012, to make the physical sense of this equality clear.

4) Eq. (5): please separate this fraction into two.

5) Section 2.1.1: you didn't mention convective adjustment scheme in CLM lake model. It should work during nights in your simulation.

6) Section 2.2: too concise description of the lake. Put more info on climate and landscape conditions, hydrological regime, previous research of the lake.

7) Line 154: "wind-only driven scheme". Again (see above), it is incorrect to state that basic CLM lake model includes only wind forcing, as it accounts for both stable and unstable stratification.

8) Section 2.3: I would add more info on the organization of measurements. Is there a mast on a lake? Which organization runs measurements? Any relevant references?

9) Line 173: "estimates a stratified lake" : sounds badly, please rephrase.

10) Table 1 is too small, you can easily present those numbers directly in text.

11) Lines 183-184: "Thermal forcing played a vital role in this enlarged diffusivity, which was considered only in CLM-KPP and not in CLM-ORG." See my comment 7 above and general comment.

12) Line 188: "$10^{-7}$" please put units and elsewhere in the document.

13) Line 188: "was the product" It is not product, but a sum.

14) Lines 198-201: two sentences, stating almost the same.

15) Line 238: "absorbed solar radiation". It is radiation flux.

16) Lines 239-240: "total eddy diffusivity". Better: total diffusivity.

17) Eq. (A3): a0, a1, ... Better to put numbers into subscript ($a_0, a_1, ...$).

18) Eq. (A4) (both equations): there is a derivative sign in numerator and not in denominator.

19) Line 244: Not clear, what is $v(h)$? You say, it is "water diffusivity". But, water

diffusivity is $K_w$. There are also molecular diffusivity, background diffusivity, diffusivity caused by internal waves . . . all denoted differently above.

20) Line 246: replace "buoyancy difference" by "buoyancy".

**References**

Subin Z.M., Riley W.J., Mironov D. An improved lake model for climate simulations: Model structure, evaluation, and sensitivity analyses in CESM1 // J. Adv. Model. Earth Syst. 2012. Vol. 4. No. 1. p. M02001.

---

## Referee Comment (RC2) · Anonymous Referee #2 · 15 Sep 2019

First, thank you for sharing your work. This is a very interesting study! You present a method of improving the thermal mixing of lakes in the Community Land Model (CLM). The new method introduced into CLM is K profile parameterization (CLM-KPP), a method utilized in ocean modeling. The current CLM vertical mixing scheme (CLM-ORG) assumes wind is the primary forcing in thermal mixing of lakes. KPP uses wind and surface thermal forcing to simulate lake temperatures. The model did not improve until a mixing event occurred on 16-31 August. CLM-ORG predicted a continued stratification of lake temperature from 16-31 August. CLM-KPP correctly estimated when and the magnitude at which the thermal mixing event would occur from 16-31 August. You provide a thorough analysis as to how thermal forcing within CLM-KPP was able

to correctly predict that the mixing would occur. However, I believe there a couple of points that would enhance this work.

Major Comments 1. The study seems limited using only one lake and a very narrow time frame. I would recommend the inclusion of several study locations and/or a longer period of analysis to get a better sense of the implications of using CLM-KPP over CLM-ORG. Right now the impact of the study feels limited given that only one location is examined for a two month period during the same season. 2. Related to 1, you do not provide an analysis of how the stratification beginning on 16 Aug better informs ecosystem, meteorological, or climatological analysis for the lake. A better discussion of implications of capturing this mixing, particularly if any were observed, would enhance this work. 3. Line 169-180: You discuss how RMSE and correlation (R) improved with CLM-KPP only slightly for the entire simulation period. I suggest that since you use these metrics, divide the calculation of these metrics into a before and after the mixing event occurs. This would strengthen your point. You should then note this in the abstract and conclusions to better illustrate the impact that CLM-KPP has in the simulation.

Minor Comments Line 100: Please define phi Line 161: How did you decided upon the 24 layers you specify?

---

## Short Comment (SC2) · 10 Oct 2019

Thanks for the authors to address my comments patiently. Overall, the response is great. Just to remind that MODIS data is probably not good for lake model validation at specific lakes, especially at the spring and fall mixing periods when the rapid change of weather would introduce significant uncertainties (such as cloud cover). Thus, the uncertainty of MODIS data need to be acknowledged. In addition, I do not think that the overestimation of surface temperature by CLM-ORG in summer is due to lack of mixing (Fig. R1). The other causes, such as the representation of latent and sensible heat, need to be acknowledged.

---

## Author Comment (AC1) · 10 Oct 2019

We thank the reviewer very much for the valuable comments on our manuscript. The comments (bolded) are addressed below.

**First, thank you for sharing your work. This is a very interesting study! You present a method of improving the thermal mixing of lakes in the Community Land Model (CLM). The new method introduced into CLM is K profile parameterization (CLM-KPP), a method utilized in ocean modeling. The current CLM vertical mixing scheme (CLMORG) assumes wind is the primary forcing in thermal mixing of lakes. KPP uses wind and surface thermal forcing to simulate lake temperatures. The model did not improve until a mixing event occurred on 16-31 August. CLM-ORG predicted a continued stratification of lake temperature from 16-31 August. CLM-KPP correctly estimated when and the magnitude at which the thermal mixing event would occur from 16-31 August. You provide a thorough analysis as to how thermal forcing within CLM-KPP was able to correctly predict that the mixing would occur. However, I believe there a couple of points that would enhance this work.**

**Major Comments 1. The study seems limited using only one lake and a very narrow time frame. I would recommend the inclusion of several study locations and/or a longer period of analysis to get a better sense of the implications of using CLM-KPP over CLM-ORG. Right now the impact of the study feels limited given that only one location is examined for a two month period during the same season.**

**Response:** We chose another lake, Nam Co, located in the Tibetan Plateau (TP) to evaluate CLM-ORG and CLM-KPP. We conducted simulations at a 10-km spatial resolution for this lake over the period of 2003 through 2012. Our simulations showed that the lake water surface temperature simulations with CLM-KPP were significantly improved when compared with observations and CLM-ORG simulations. We have added simulations and analysis for Nam Co to the manuscript:

"We validated both CLM-ORG and CLM-KPP with the monthly Moderate Resolution Imaging Spectroradiometer (MODIS) data for Nam Co by conducting 10-km spatial resolution simulations for this lake over the period of 2003 through 2012. We can see that CLM-KPP improved WST simulations averaged over the entire lake (34 model grid cells) when compared with the MODIS data and CLM-ORG simulations (Fig. R1). The RMSE of WST decreased from 4.58 ˚C with CLM-ORG to 2.23 ˚C with CLM-KPP, and the R increased from 0.90 to 0.96 at the same time.

The improved WST simulations with CLM-KPP were closely related to the water mixing simulations with the KPP as discussed in the manuscript. We averaged the $K_w^{ORG}$ and $K_w^{KPP}$ simulations over the water columns with the depth greater than 25 m for Nam Co as shown in Fig. R2, and the total of such columns were 28 out of 34 for this lake. Figure R2 indicated that $K_w^{KPP}$ was slightly smaller than $K_w^{ORG}$ mostly in the mixing layer of the lake over the summer. The difference likely resulted from the enlarged $K_w^{ORG}$ in CLM where this parameter was increased by a factor of 10 when the lake depth was greater than 25 m. In the deeper part of the lake, $K_w^{KPP}$ was much smaller than $K_w^{ORG}$ over the summer due much to

$K_{ed}$'s contribution to $K_w^{ORG}$. In the spring and fall seasons, $K_w^{KPP}$ was significantly larger than $K_w^{ORG}$ where the buoyancy flux may contribute strongly to $K_w^{KPP}$. During the winter time when the lake froze, both CLM-KPP and CLM-ORG were set to use $K_w^{ORG}$. We can see that the most significant improvements in WST for Nam Co occurred during the ice-free seasons when the KPP was activated. Thus, the thermal forcing was an important factor in simulating lake mixing, which needs to be considered in lake models."

[Figure]

Figure R1. The time series over the period of 2003 through 2012 of monthly WST observations from MODIS (black star line) and simulations with CLM-ORG (blue line) and CLM-KPP (red line) (Unit: ˚C).

[Figure]

Figure R2. The simulated (a) $\log_{10} K_w^{ORG}$ with CLM-ORG, (b) $\log_{10} K_w^{KPP}$ with CLM-KPP (Unit: m²/s) averaged over the water columns with the depth greater than 25 m (28 of 34 grid cells), and (c) the differences between $\log_{10} K_w^{KPP}$ and $\log_{10} K_w^{ORG}$ ($\log_{10} K_w^{KPP} - \log_{10} K_w^{ORG}$).

**2. Related to 1, you do not provide an analysis of how the stratification beginning on 16 Aug better informs ecosystem,**

**meteorological, or climatological analysis for the lake. A better discussion of implications of capturing this mixing, particularly if any were observed, would enhance this work.**

**Response:** Stratification plays an important role in lake production and food webs. Stratification and warmer epilimnion temperatures create conditions necessary for phytoplankton production. Also, when arctic lakes become strongly stratified, the hypolimnion can become anoxic, which in turn, increases nutrient recycling and leads to elevated production the following spring (O'Brien et al., 2005). Increased food availability and warmer lake temperatures in the epilimnion from stratification increase arctic char growth. Finally, simulations of stratification date and epilimnion temperature are used in bioenergetic models to estimate fish growth and consumption and better understand arctic char production with global environmental change (Budy and Luecke, 2014).

**3. Line 169-180: You discuss how RMSE and correlation (R) improved with CLM-KPP only slightly for the entire simulation period. I suggest that since you use these metrics, divide the calculation of these metrics into a before and after the mixing event occurs. This would strengthen your point. You should then note this in the abstract and conclusions to better illustrate the impact that CLM-KPP has in the simulation.**

**Response:** Based on this comment, we separated our entire simulation period for Fog3 Lake into the before and after mixing periods and calculated RMSEs and Rs for these two periods as well as the entire simulation period (Table R1). We can see that CLM-KPP remarkably improved the water mixing simulations in Fog3 Lake when compared with CLM-ORG.

Table R1. RMSEs (°C) and Rs of the temperature profile simulations with CLM-ORG and CLM-KPP for Fog3 Lake for the three periods of 1 July through 15 August, 16–31 August, and 1 July through 31 August in 2018.

| | 1 July–15 August, 2018 | | 16–31 August, 2018 | | 1 July–31 August, 2018 | |
|---|---|---|---|---|---|---|
| | RMSE (°C) | R | RMSE (°C) | R | RMSE (°C) | R |
| CLM-ORG | 1.1 | 0.93 | 1.4 | 0.57 | 1.2 | 0.90 |
| CLM-KPP | 1.3 | 0.92 | 0.3 | 0.99 | 1.0 | 0.95 |

**Minor comments**

**Line 100: Please define phi.**

**Response:** Actually, we define phi ($\varphi$) in Line 99 "$k^*$ is related to latitude $\varphi$" in the manuscript.

**Line 161: How did you decided upon the 24 layers you specify?**

**Response:** Observed lake temperatures for Fog3 Lake are for the lake depths of 0, 1, 2, 3, 4, 5, 6, 7, 8, 10, 12, 14, and 16 m. CLM-ORG has 10 lake layers by default, and the center point depths of these layers are 0.05, 0.3, 0.9, 1.9, 3.3, 5.1, 7.5, 10.3,

13.79, and 17.94 m generated automatically by the layering scheme in the model based on the input lake depth. For this study, we tried to keep each layer thin in the top part of the lake to reflect diurnal cycles (layers 1-5) in both CLM-ORG and CLM-KPP. Below layer 5, mostly we used the observed points to layer the rest of the lake column. Finally, we produced 24 layers for the entire lake column in both models, and the center point depths of these lake layers are 0.05, 0.15, 0.25, 0.35, 0.45, 1, 2, 3, 4, 5, 6, 7, 8, 9, 10, 11, 12, 13, 14, 15, 16, 17, 18, and 19.25 m, respectively. As shown in the Fig. R3 and Table R2, the simulations of CLM-ORG with both 10 and 24 layers were very similar, while the simulations of CLM-KPP with 24 layers were closer to observations than those with 10 layers when the water mixing event occurred. These results indicated that a high resolution layering in CLM-KPP may be important for simulating lake water mixing.

[Figure]

Figure R3. Lake temperature profiles of (a) observations, simulations of (b) CLM-ORG and (c) CLM-KPP with 24 layers and simulations of (d) CLM-ORG and (e) CLM-KPP with 10 layers (unit: ˚C).

Table R2. RMSEs (˚C) and Rs of the temperature profile simulations of CLM-ORG and CLM-KPP with 10 and 24 layers for Fog3 Lake over the three periods of 1 July through 15 August, 16–31 August, and 1 July through 31 August in 2018.

| | | 1 July–15 August, 2018 | | 16–31 August, 2018 | | 1 July–31 August, 2018 | |
|---|---|---|---|---|---|---|---|
| | | RMSE (˚C) | R | RMSE (˚C) | R | RMSE (˚C) | R |
| 10 model | CLM-ORG | 1.0 | 0.94 | 1.4 | 0.58 | 1.1 | 0.92 |
| layers | CLM-KPP | 1.2 | 0.94 | 0.7 | 0.90 | 1.1 | 0.94 |
| 24 model | CLM-ORG | 1.1 | 0.93 | 1.4 | 0.57 | 1.2 | 0.90 |
| layers | CLM-KPP | 1.3 | 0.92 | 0.3 | 0.99 | 1.0 | 0.95 |

References

Budy, P., Luecke, C.: Understanding how lake populations of arctic char are structured and function with special consideration of the potential effects of climate change: a multi-faceted approach, Oecologia, 176(1), 81-94, https://dx.doi.org/10.1007/s00442-014-2993-8, 2014.

O'Brien, W.J., Barfield, M., Bettez, N., Hershey, A.E., Hobbie, J.E., Kipphut, G., Kling, G., Miller, M.C.: Long-term response and recovery to nutrient addition of a partitioned arctic lake, Freshwater. Biol., 50(5), 731-741, https://doi.org/10.1111/j.1365-2427.2005.01354.x, 2005.

---

## Author Comment (AC2) · 10 Oct 2019

We thank the reviewer very much for the valuable comments on our manuscript. The comments (bolded) are fully addressed as follows.

**The manuscript presents results of introducing "K profile" parameterizatioin of turbulence into lake module of Community Land Model. This is likely the first time K profile parameterization is used in a 1D lake model, though it is widely applied in ocean models. Incorporation of new turbulence closure instead of standard Henderson-Seller diffusivity lead to significant improvement of simulation of late-summer destratification event in an Alaskan lake.**

**General comment**

**My general comment on the manuscript is that since single mixing event is simulated, more physical analysis could be provided to explain why K profile closure performed better than Henderson-Sellers in this case. Analysis presented in sections 3.1 and 3.2 is superficial and does not touch this question. One mixing case is not enough to state that K profile is better in similar situations in general, so more substantial inquiry into physics behind both parameterizations is needed. The authors state that KPP includes effects of thermal forcing, whereas original scheme of CLM model does not. This is actually not correct. First, original CLM model includes convective adjustment scheme (Subin et al., 2012) which instantaneously mixes the unstably stratified water column. Then, the effects of stable stratification are included via Brunt-Vaisala frequency in Henderson-Sellers (H-S) diffusivity. Thus, thermal (density) stratification is taken into account. The mixing event the authors focus on happens during weakly stable stratification under strong wind forcing. One may conclude from simulation results presented is that given the same stable temperature profile the larger wind speed is needed for H-S to mix completely the water column than for KPP model. This may be elaborated by conducting idealized simulations with both turbulence closures with varying wind speeds and temperature profiles where this statement may be checked and respective quantitative estimates provided.**

**Response:** Thank you for your insightful comments. We modified several places in the manuscript to address your questions. Our general reply is as follows.

The difference of the current mixing parameterization of the CLM (CLM-ORG) and the KPP (CLM-KPP) is in the equations used to estimate eddy diffusivity. In CLM-KPP, the eddy diffusivity is estimated separately for the lake boundary layer and lake interior. In the lake boundary layer, the eddy diffusivity is not determined by local gradient of mean variables, but it is determined by surface forcing and the boundary layer depth. The non-local effect is taken into account by estimating the boundary layer depth first, and the eddy diffusivity is specified with a prescribed profile in the boundary layer. In the lake interior, the mixing is generally weak and associated with internal wave activity and shear instability. From our point of view, the major shortcomings of CLM-ORG are that it does not consider a boundary layer for eddy development, and it requires an

ad hoc parameter to enhance the estimated eddy diffusivity. In the KPP scheme, an explicit inclusion of an ad hoc enlarging parameter is avoided. The KPP scheme was tested for different time scales, diurnal change, seasonal cycle, and single event for different locations (Large et al. 1994). We have also conducted more simulations for other lakes with quite different environment settings, e.g. Nam Co at Tibetan Plateau with a focus on its long term change, and the results are presented below.

For Nam Co, located in the Tibetan Plateau, we conducted simulations at a 10-km spatial resolution over the period of 2003 through 2012. Our simulations showed that the lake water surface temperature simulations with CLM-KPP were significantly improved when compared with CLM-ORG simulations. We have added simulations and analysis for Nam Co to the manuscript:

"We validated both CLM-ORG and CLM-KPP with the monthly Moderate Resolution Imaging Spectroradiometer (MODIS) data for Nam Co by conducting 10-km spatial resolution simulations for this lake over the period of 2003 through 2012. We can see that CLM-KPP improved WST simulations averaged over the entire lake (34 model grid cells) when compared with the CLM-ORG simulations (Fig. R1). The RMSE of WST decreased from 4.58 ˚C with CLM-ORG to 2.23 ˚C with CLM-KPP, and the R increased from 0.90 to 0.96 at the same time.

The differences in the mixing coefficients of CLM-KPP and CLM-ORG cause the difference in WST simulations. We averaged the $K_w^{ORG}$ and $K_w^{KPP}$ over the water columns with the depth greater than 25 m for Nam Co (Fig. R2), and the total of such columns were 28 out of 34 for this lake. Figure R2 indicated that $K_w^{KPP}$ was slightly smaller than $K_w^{ORG}$ mostly in the mixed layer of the lake during summer time. In the deeper part of the lake, $K_w^{KPP}$ was much smaller than $K_w^{ORG}$ during summer time. In the spring and fall seasons, $K_w^{KPP}$ was significantly larger than $K_w^{ORG}$ where the buoyancy flux may contribute strongly to $K_w^{KPP}$. During the winter time when the lake froze, both CLM-KPP and CLM-ORG were set to use $K_w^{ORG}$. We can see that the most significant improvements in WST for Nam Co occurred during the ice-free seasons when the KPP was activated."

[Figure]

Figure R1. The time series over the period of 2003 through 2012 of monthly WST observations from MODIS (black star line) and simulations with CLM-ORG (blue line) and CLM-KPP (red line) (Unit: ˚C).

[Figure]

Figure R2. The simulated (a) $\log_{10} K_w^{ORG}$ with CLM-ORG, (b) $\log_{10} K_w^{KPP}$ with CLM-KPP (Unit: m²/s) averaged over the water columns with the depth greater than 25 m (28 of 34 grid cells), and (c) the differences between $\log_{10} K_w^{KPP}$ and $\log_{10} K_w^{ORG}$ ($\log_{10} K_w^{KPP} - \log_{10} K_w^{ORG}$).

In CLM-KPP, the eddy diffusivity formulation is different for the boundary layer and lake interior. In the lake boundary layer, the eddy diffusivity is related with boundary layer depth and surface forcing. In the lake interior, the eddy diffusivity is relatively weak, associated with internal wave activity and shear instability. Overall the CLM-KPP can enhance the eddy diffusivity during spring and fall and maintain weak eddy diffusivity in the lake interior during summer when stratification is strong. The outcome of the CLM-KPP eddy diffusivity is an improved WST simulation.

For Fog3 Lake in Alaska, numerical experiments were conducted for CLM-ORG with enhanced wind. Figures R3 and R4 showed simply providing larger winds could not significantly improve CLM-ORG simulations for this lake (Table R1). When stronger wind is used, the CLM-ORG can simulate the mixing event around 16 Aug. However, the strong wind causes WST to have a negative bias, presumably caused by heat loss from the lake. Thus, as shown in the manuscript, the CLM-KPP provides a better parameterization of eddy diffusivity and improved lake temperature simulations.

[Figure]

Figure R3. WST observations (black line) and CLM-ORG simulations with the default wind data (red line), with wind data 2-fold increased (blue line), with wind data 1.5-fold increased (green line), and with wind data 1.8-fold increased (grey line) (unit: ˚C).

[Figure]

Figure R4. Lake temperature profiles of (a) observations and CLM-ORG simulations with (b) the default wind data, (c) with wind data 2-fold increased, (d) with wind data 1.5-fold increased (green line), and (e) with wind data 1.8-fold increased (grey line) (unit: ˚C).

Table R1. RMSEs (˚C) and Rs of the temperature profile simulations with CLM-ORG, the case with wind 2-fold increased, the case with wind 1.5-fold increased, and the case with wind 1.8-fold increased for Fog3 Lake for the three periods of 1 July through 15 August, 16−31 August, and 1 July through 31 August in 2018.

|  | 1 July−15 August, 2018 | | 16−31 August, 2018 | | 1 July−31 August, 2018 | |
| --- | --- | --- | --- | --- | --- | --- |
|  | RMSE (˚C) | R | RMSE (˚C) | R | RMSE (˚C) | R |
| CLM-ORG | 1.1 | 0.93 | 1.4 | 0.57 | 1.2 | 0.90 |
| wind×2 | 2.0 | 0.83 | 0.5 | 0.63 | 1.7 | 0.83 |
| wind×1.5 | 0.83 | 0.97 | 1.1 | 0.59 | 0.91 | 0.94 |
| wind×1.8 | 2.0 | 0.84 | 0.6 | 0.62 | 1.6 | 0.84 |

We mentioned the convective adjustment scheme in the manuscript. The convection scheme works when there exists density

instability (Hostetler and Bartlein, 1990).

**Specific comments**

**(1) Lines 88-90: "Researchers have attempted to advance this lake model to more closely reflect reality over the last two decades (Fang and Stefan, 1996; Henderson-Sellers, 1985; Hostetler and Bartlein, 1990; Subin et al., 2012)." Three of four papers cited here do not deal with CLM model.**

**Response:** We deleted this sentence.

**(2) Lines 92-93: "... is the enhanced eddy diffusivity for unresolved mixing processes". All mixing processes in 1D model are unresolved and are parameterized, because only 3D model of sufficiently high resolution simulates turbulence explicitly.**

**Response:** We agreed with this reviewer on this comment. We changed "for unresolved mixing processes" to "to strengthen mixing processes."

**(3) Line 98: "$0.0012u_2$" I guess, you can write drag coefficient $C_d$ instead of 0.0012, to make the physical sense of this equality clear.**

**Response:** Yes, we changed "0.0012" to "$C_d$" in the manuscript.

**(4) Eq. (5): please separate this fraction into two.**

**Response:** Yes, we separated the fraction into two parts.

**(5) Section 2.1.1: you didn't mention convective adjustment scheme in CLM lake model. It should work during nights in your simulation.**

**Response:** Yes, based on the general comment, we included convective adjustment scheme to the manuscript.

**(6) Section 2.2: too concise description of the lake. Put more info on climate and landscape conditions, hydrological regime, previous research of the lake.**

**Response:** Yes, we added more description of Fog3 Lake to the manuscript:

Change "Fog 3 Lake, is in Arctic Alaska at (68.67˚ N, 149.10˚ W) (Fig. 1a). In 2018 it had a surface area of 35,230 m$^2$ and a maximum depth of 19.74 m. The lake has a long ice duration, and ice-off is usually in late June, while ice-on typically occurs in early October (Arp et al., 2015)."

to

"Fog 3 Lake, is in Arctic Alaska at (68.67˚ N, 149.10˚ W) (Fig. 1a). In 2018 it had a surface area of 38,863 m$^2$ and a maximum depth of 20.96 m. The lake has a long ice duration, and ice-off is usually in late June, while ice-on typically occurs in early October (Arp et al., 2015). Around this lake, the mean annual air temperature is about ~ -6 ℃, and the mean annual precipitation is ~ 200 mm (Ping et al., 1998). This kettle lake is surrounded with lower hills mainly covered with shrubs and tundra. Due to the treeless and landscape, there are no effects from tree shielding on wind. In addition, Fog 3 Lake is formed by glaciers, and has less connection to other surrounding surface waters."

**(7) Line 154: "wind-only driven scheme". Again (see above), it is incorrect to state that basic CLM lake model includes only wind forcing, as it accounts for both stable and unstable stratification.**

**Response:** Yes, see the response for the general comment.

**(8) Section 2.3: I would add more info on the organization of measurements. Is there a mast on a lake? Which organization runs measurements? Any relevant references?**

**Response:** Fog3 Lake is near Toolik Field Station (68°37.796' N, 149°35.834' W), in the northern foothills of the Brooks Mountain Range, Alaska (https://toolik.alaska.edu/edc/abiotic_monitoring/index.php). The weather station is on the shore of Fog 3 Lake, and Utah State University runs the measurements included in this study.

**(9) Line 173: "estimates a stratified lake": sounds badly, please rephrase.**

**Response:** Yes, we changed this sentence to **"CLM-KPP accurately captured the mixing event (Fig. 3c), while CLM-ORG produced strong stratification in the upper part of the lake throughout the simulation period (Fig. 3b)."**

**(10) Table 1 is too small, you can easily present those numbers directly in text.**

**Response:** We separated our entire simulation period for Fog3 Lake into the before and after mixing periods and calculated RMSEs and Rs for these two periods as well as the entire simulation period (Table R2). We can see that CLM-KPP remarkably improved the water mixing simulations in Fog3 Lake when compared with CLM-ORG.

Table R2. RMSEs (˚C) and Rs of the temperature profile simulations with CLM-ORG and CLM-KPP for Fog3 Lake for the three periods of 1 July through 15 August, 16–31 August, and 1 July through 31 August in 2018.

| | 1 July–15 August, 2018 | | 16–31 August, 2018 | | 1 July–31 August, 2018 | |
|---|---|---|---|---|---|---|
| | RMSE (˚C) | R | RMSE (˚C) | R | RMSE (˚C) | R |
| CLM-ORG | 1.1 | 0.93 | 1.4 | 0.57 | 1.2 | 0.90 |
| CLM-KPP | 1.3 | 0.92 | 0.3 | 0.99 | 1.0 | 0.95 |

**(11) Lines 183-184: "Thermal forcing played a vital role in this enlarged diffusivity, which was considered only in CLM-KPP and not in CLM-ORG." See my comment 7 above and general comment.**

**Response:** Yes, see the response for the general comment.

**(12) Line 188: "$10^{-7}$" please put units and elsewhere in the document.**

**Response:** Yes, we put units and elsewhere in the manuscript.

**(13) Line 188: "was the product" It is not product, but a sum.**

**Response:** Yes, we changed "product" to "sum" in the manuscript.

**(14) Lines 198-201: two sentences, stating almost the same.**

**Response:** The first sentence states the $N^2$, while the second sentence states the water stratification.

**(15) Line 238: "absorbed solar radiation". It is radiation flux.**

**Response:** Yes, we modified "absorbed solar radiation" to "radiation flux" in the manuscript.

**(16) Lines 239-240: "total eddy diffusivity". Better: total diffusivity.**

**Response:** Yes, we modified "total eddy diffusivity" to "total diffusivity" in the manuscript.

**(17) Eq. (A3): a0, a1, ... Better to put numbers into subscript ($a_0$, $a_1$, ...).**

**Response:** Yes, we put numbers into subscript accordingly in the manuscript.

**(18) Eq. (A4) (both equations): there is a derivative sign in numerator and not in denominator.**

**Response:** Yes, we made it more clearly in the manuscript.

**(19) Line 244: Not clear, what is $\vartheta(h)$? You say, it is "water diffusivity". But, water diffusivity is $K_w$. There are also molecular diffusivity, background diffusivity, diffusivity caused by internal waves … all denoted differently above.**

**Response:** $\vartheta(h)$ refers to the total diffusivity of water, a sum of molecular diffusivity, background diffusivity, diffusivity caused by internal waves. We made it more clearly in the manuscript.

**(20) Line 246: replace "buoyancy difference" by "buoyancy".**

**Response:** Yes, we replaced "buoyancy difference" by "buoyancy" in the manuscript.

References

Arp, C. D., Jones, B. M., Liljedahl, A. K., Hinkel, K. M., and Welker, J. A.: Depth, ice thickness, and ice-out timing cause divergent hydrologic responses among Arctic lakes, Water Resour. Res., 51(12), 9379-9401, https://dx.doi.org/10.1002/2015WR017362, 2015.

Hostetler, S. W., and Bartlein, P. J.: Simulation of lake evaporation with application to modeling lake level variations of Harney-Malheur Lake, Oregon, Water Resour. Res., 26(10), 2603-2612, https://dx.doi.org/10.1029/WR026i010p02603, 1990.

Large, W. G., Mcwilliams, J. C., and Doney, S. C.: Oceanic vertical mixing: A review and a model with a nonlocal boundary layer parameterization, Rev. Geophys., 32(4), 363-403, https://dx.doi.org/10.1029/94RG01872, 1994.

Ping, C. L., Bockheim, J. G., Kimble, J. M., Michaelson, G. J., and Walker, D. A: Characteristics of cryogenic soils along a latitudinal transect in arctic Alaska, J. Geophys. Res., 103, 28917-28928, https://dx.doi.org/10.1029/98JD02024, 1998.

---

## Author Comment (AC3) · 10 Oct 2019

We thank you very much for the constructive and helpful comments on our manuscript. The comments (bolded) from the reviewer Dr. Zeli Tan are fully addressed in the following.

**It is an interesting study. Because a 1-D lake model is still much needed to understand the impact of climate changes on global lake systems, a parameterization method that could improve the simulation of lake mixing process will be much valued. But I suggest that the manuscript can be improved in the following directions. First of all, the comparison between CLM-ORG and CLM-KPP is not exhausted, to day the least. In Subin's CLM-ORG paper, he actually tested the model over a pair of lakes around the globe. In fact, the CLM-ORG performance on high-latitude lakes which this study focused on was not the worst. Thus, the method can become much more valuable if the authors can apply this method to some more lakes, especially those deep and large lakes.**

**Response:** Thanks for the comments. We selected one additional lake, Nam Co, located in the Tibetan Plateau (TP) to validate CLM-ORG and CLM-KPP. Nam Co is the largest lake in the central TP with a surface area of about 2,021 km$^2$ in 2010 (Lei et al., 2013; Zhu et al., 2010). Its maximum depth reaches more than 95 m, and the mean depth is about 40 m (Wang et al., 2009). Thus, we chose this large and deep lake to evaluate our newly coupled model. We have added simulations and analysis for Nam Co to the manuscript. The results are described as follows:

"We validated both CLM-ORG and CLM-KPP with the monthly Moderate Resolution Imaging Spectroradiometer (MODIS) data for Nam Co by conducting 10-km spatial resolution simulations for this lake over the period of 2003 through 2012. We can see that CLM-KPP improved WST simulations averaged over the entire lake (34 model grid cells) when compared with the MODIS data and CLM-ORG simulations (Fig. R1). The RMSE of WST decreased from 4.58 ˚C with CLM-ORG to 2.23 ˚C with CLM-KPP, and the R increased from 0.90 to 0.96 at the same time.

The improved WST simulations with CLM-KPP were closely related to the water mixing simulations with the KPP as discussed in the manuscript. We averaged the $K_w^{ORG}$ and $K_w^{KPP}$ simulations over the water columns with the depth greater than 25 m for Nam Co as shown in Fig. R2, and the total of such columns were 28 out of 34 for this lake. Figure R2 indicated that $K_w^{KPP}$ was slightly smaller than $K_w^{ORG}$ mostly in the mixing layer of the lake over the summer. The difference likely resulted from the enlarged $K_w^{ORG}$ in CLM where this parameter was increased by a factor of 10 when the lake depth was greater than 25 m. In the deeper part of the lake, $K_w^{KPP}$ was much smaller than $K_w^{ORG}$ over the summer due much to $K_{ed}$'s contribution to $K_w^{ORG}$. In the spring and fall seasons, $K_w^{KPP}$ was significantly larger than $K_w^{ORG}$ where the buoyancy flux may contribute strongly to $K_w^{KPP}$. During the winter time when the lake froze, both CLM-KPP and CLM-ORG were set to use $K_w^{ORG}$. We can see that the most significant improvements in WST for Nam Co occurred during the ice-free seasons when the KPP was activated. Thus, the thermal forcing was an important factor in simulating lake mixing, which needs to be considered in lake models."

[Figure]

Figure R1. The time series over the period of 2003 through 2012 of monthly WST observations from MODIS (black star line) and simulations with CLM-ORG (blue line) and CLM-KPP (red line) (Unit: ˚C).

[Figure]

Figure R2. The simulated (a) $\log_{10} K_w^{ORG}$ with CLM-ORG, (b) $\log_{10} K_w^{KPP}$ with CLM-KPP (Unit: m$^2$/s) averaged over the water columns with the depth greater than 25 m (28 of 34 grid cells), and (c) the differences between $\log_{10} K_w^{KPP}$ and $\log_{10} K_w^{ORG}$ ($\log_{10} K_w^{KPP} - \log_{10} K_w^{ORG}$).

**Second, more information about the study lake is needed. Is Fog3 Lake a glacial lake or a thermokarst lake? How was the surface friction velocity derived for this lake? Are the effects of lake fetch and wind shielding considered?**

**What is the lake's light attenuation coefficient?**

**Response:** Fog3 Lake is a glacial lake. In CLM-ORG, the surface friction velocity $w^*$ (m/s) is calculated as:

$$w^* = 0.0012u_2 \tag{R1}$$

where $u_2$ is the 2-m wind speed (m/s).

While in CLM-KPP, the surface friction velocity $u^*$ (m/s) is calculated as (Large and Pond, 1982):

$$u^{*2} = \frac{\rho_a}{\rho} C_D U^2 \tag{R2a}$$

$$10^3 C_D = \frac{2.70}{U} + 0.142 + 0.0764U \tag{R2b}$$

where $\rho_a$ and $\rho$ are the air and lake water densities (kg/m$^3$) respectively, $C_D$ is the drag coefficient and $U$ is the 10-m wind speed (m/s). The effect of the lake fetch was considered in our simulations. In the CLM-ORG, the lake fetch F (m) (Hutchinson, 1957; Wetzel and Likens, 1991) is:

$$F = \begin{cases} 100, & D < 4 \\ 25D, & D \geq 4 \end{cases} \tag{R3}$$

where D is the water depth. We also used this function in CLM-KPP.

In this study, wind shielding was not considered. Actually, the Toolik meteorological station providing the wind data is ~1.5 km away from Fog3 Lake, although there are no buildings or trees between the Toolik station and the lake. Thus, the wind shielding effects are not significant. The light extinction coefficient η (m$^{-1}$) is a function of depth (m) (Hakanson, 1995):

$$\eta = 1.1925D^{-0.424} \tag{R4}$$

In this study, with the lake depth (D) of 20 m for Fog3 Lake, η is about 0.33 m$^{-1}$.

**Third, how are CLM-ORG and CLM-KPP calibrated in this study? I know that CLM-ORG has a water mixing parameter that can be used to increase diffusivity for those deep lakes. Can the parameter values of CLM-KPP described here be applied to other lakes?**

**Response:** Both CLM-ORG and CLM-KPP were not calibrated in this study. Yes, the water mixing parameter in CLM-ORG can be increased to generate stronger water mixing for deep lakes (Gu et al., 2013). Here, we increased the water diffusivity (Eq. (1) in the manuscript) by 10 and 100 times in CLM-ORG and conducted additional simulations for Fog3 Lake as shown in Figs. R3 and R4. We can see that CLM-ORG was still unable to reproduce the observed lake temperatures with the enlarged water diffusivity. Again, we did not adjust any parameters in CLM-KPP when we performed simulations for Fog3 Lake, and the same parameters were applied to the simulations for Nam Co. We see that CLM-KPP more realistically captured the water mixing in Nam Co than CLM-ORG (Figs. R1 and R2).

[Figure]

Figure R3. Lake water surface temperature observations (black line), simulations with CLM-ORG (red line), and simulations with $K_w^{ORG}$ multiplied by 10 (blue line) and 100 (green line), respectively.

[Figure]

Figure R4. Lake temperature profiles of (a) observations, (b) simulations with CLM-ORG, and simulations with $K_w^{ORG}$ multiplied by (c) 10 and (d) 100.

**Forth, I am surprised that the case study did not cover the period of spring water mixing which can have large biogeochemical impacts for high-latitude lakes.**

**Response:** Lake temperature data and some of the atmospheric forcing data for Fog3 Lake are available only for July and August 2018. However, our additional simulations with CLM-ORG and CLM-KPP for Nam Co covered the period of 2003-2012, which included the spring season (Figs. R1 and R2). Our simulations with CLM-KPP were closer to observations than those with CLM-ORG for almost the entire simulation period including the spring seasons.

References

Gu, H., Jin, J., Wu, Y., Ek, M. B., and Subin, Z. M.: Calibration and validation of lake surface temperature simulations with the coupled WRF-lake model, Clim. Change, 129(3-4), 471-483, https://dx.doi.org/10.1007/s10584-013-0978-y, 2013.

Hakanson, L.: Models to predict Secchi depth in small glacial lakes, Aquat Sci, 57, 31-53, https://doi.org/10.1007/BF00878025, 1995.

Hutchinson, G. E.: A treatise on Limnology, vol. 1, Geography, Physics, and Chemsitry, John Weiley, New York, 1957.

Large, W. G. and Pond, S.: Sensible and latent heat flux measurements over the ocean, J. Phys. Oceanogr., 12, 464-482, https://doi.org/10.1175/1520-0485(1982)012<0464:SALHFM>2.0.CO;2, 1982.

Lei, Y., Yao, T., Bird, B. W., Yang, K., Zhai, J., and Sheng, Y.: Coherent lake growth on the central Tibetan Plateau since the 1970s: Characterization and attribution, J. Hydrol., 483, 61-67, https://dx.doi.org/10.1016/j.jhydrol.2013.01.003, 2013.

Wang, J., Zhu, L., Daut, G., Ju, J., Lin, X., Wang, Y., and Zhen, X.: Investigation of bathymetry and water quality of Lake Nam Co, the largest lake on the central Tibetan Plateau, China, Limnology, 10(2), 149-158, https://dx.doi.org/10.1007/s10201-009-0266-8, 2009.

Wetzel, R., and Likens, G. E.: Limnological Analyses, Springer, New York, 1991.

Zhu, L., Xie, M., and Wu, Y.: Quantitative analysis of lake area variations and the influence factors from 1971 to 2004 in the Nam Co basin of the Tibetan Plateau, Chin. Sci. Bull., 55(13), 1294-1303, https://dx.doi.org/10.1007/s11434-010-0015-8, 2010.

---

## Author Comment (AC4) · 16 Oct 2019

Thank you very much for the comments. The comments (bolded) from the reviewer Dr. Zeli Tan are fully addressed in the following.

**Thanks for the authors to address my comments patiently. Overall, the response is great.**

**Just to remind that MODIS data is probably not good for lake model validation at specific lakes, especially at the spring and fall mixing periods when the rapid change of weather would introduce significant uncertainties (such as cloud cover). Thus, the uncertainty of MODIS data need to be acknowledged.**

**Responses:** Yes, we will acknowledge the uncertainties of MODIS data in the manuscript. Previous studies have evaluated MODIS water surface temperature data for lakes based on *in situ* observations (Crosman and Horel, 2009; Schneider et al., 2009). Zhang et al. (2014) compared the nighttime water surface temperature of MODIS and *in situ* observations for Nam Co with a correlation coefficient of 0.89 and a bias of -1.4 ℃, which showed an acceptable accuracy for MODIS. Thus, due to limited observed temperature data for Nam Co, we chose MODIS data to validate CLM-ORG and CLM-KPP.

**In addition, I do not think that the overestimation of surface temperature by CLM-ORG in summer is due to lack of mixing (Fig. R1). The other causes, such as the representation of latent and sensible heat, need to be acknowledged.**

**Responses:** Yes, Figure R2 showed that $K_w^{KPP}$ was slightly smaller than $K_w^{ORG}$ mostly in the mixed layer of the lake during summer time. In the deeper part of the lake, $K_w^{KPP}$ was much smaller than $K_w^{ORG}$ during summer time. In the spring and fall seasons, $K_w^{KPP}$ was significantly larger than $K_w^{ORG}$ (see the previous responses to Dr. Zeli Tan in detail).

In CLM-KPP, the eddy diffusivity formulation is different for the boundary layer and lake interior. In the lake boundary layer, the eddy diffusivity is related with boundary layer depth and surface forcing. In the lake interior, the eddy diffusivity is relatively weak, associated with internal wave activity and shear instability. Overall, CLM-KPP enhances the eddy diffusivity during the spring and fall and maintains a weak eddy diffusivity in the lake interior during the summer when stratification is strong when compared to CLM-ORG. In addition, the overestimated water surface temperature with CLM-ORG before the summer affects the energy budget at the lake surface, which further influences lake temperature simulations during the summer.

[Figure]

Figure R1. The time series over the period of 2003 through 2012 of monthly WST observations from MODIS (black star line) and simulations with CLM-ORG (blue line) and CLM-KPP (red line) (Unit: °C).

[Figure]

Figure R2. The simulated (a) $\log_{10} K_w^{ORG}$ with CLM-ORG, (b) $\log_{10} K_w^{KPP}$ with CLM-KPP (Unit: $m^2$/s) averaged over the water columns with the depth greater than 25 m (28 of 34 grid cells), and (c) the differences between $\log_{10} K_w^{KPP}$ and $\log_{10} K_w^{ORG}$ ($\log_{10} K_w^{KPP} - \log_{10} K_w^{ORG}$).

References

Crosman, E. T., and Horel, J. D.: MODIS-derived surface temperature of the Great Salt Lake, Remote Sens. Environ., 113(1), 73-81, https://dx.doi.org/10.1016/j.rse.2008.08.013, 2009.

Schneider, P., Hook, S. J., Radocinski, R. G., Corlett, G. K., Hulley, G. C., Schladow, S. G., and Steissberg, T. E.: Satellite observations indicate rapid warming trend for lakes in California and Nevada, Geophys. Res. Lett., 36, L22402, https://dx.doi.org/10.1029/2009GL040846, 2009.

Zhang, G., Yao, T., Xie, H., Qin, J., Ye, Q., Dai, Y., and Guo, R.: Estimating surface temperature changes of lakes in the Tibetan Plateau using MODIS LST data, 119, 8552-8567, https://dx.doi.org/10.1002/2014JD021615, 2014.

---

## Author Response (AR1)

Thanks for the editor and reviewers for the comments to improve the quality of our manuscript. We have carefully addressed the comments with point-by-point replies to the reviewers and also Dr. Zeli Tan and revised our manuscript accordingly. Attached is a marked-up version of the manuscript.

**Responses to the reviewer #1:**

We thank the reviewer #1 very much for the valuable comments on our manuscript. The comments (bolded) are fully addressed as follows.

**The manuscript presents results of introducing "K profile" parameterizatioin of turbulence into lake module of Community Land Model. This is likely the first time K profile parameterization is used in a 1D lake model, though it is widely applied in ocean models. Incorporation of new turbulence closure instead of standard Henderson-Seller diffusivity lead to significant improvement of simulation of late-summer destratification event in an Alaskan lake.**

**General comment**

**My general comment on the manuscript is that since single mixing event is simulated, more physical analysis could be provided to explain why K profile closure performed better than Henderson-Sellers in this case. Analysis presented in sections 3.1 and 3.2 is superficial and does not touch this question. One mixing case is not enough to state that K profile is better in similar situations in general, so more substantial inquiry into physics behind both parameterizations is needed. The authors state that KPP includes effects of thermal forcing, whereas original scheme of CLM model does not. This is actually not correct. First, original CLM model includes convective adjustment scheme (Subin et al., 2012) which instantaneously mixes the unstably stratified water column. Then, the effects of stable stratification are included via Brunt-Vaisala frequency in Henderson-Sellers (H-S) diffusivity. Thus, thermal (density) stratification is taken into account. The mixing event the authors focus on happens during weakly stable stratification under strong wind forcing. One may conclude from simulation results presented is that given the same stable temperature profile the larger wind speed is needed for H-S to mix completely the water column than for KPP model. This may be elaborated by conducting idealized simulations with both turbulence closures with varying wind speeds and temperature profiles where this statement may be checked and respective quantitative estimates provided.**

**Response:** Thank you for the insightful comments. We modified several places in the manuscript to address your questions. Our general reply is as follows.

The difference of the current mixing parameterization of the CLM (CLM-ORG) and the KPP (CLM-KPP) is in the equations used to estimate eddy diffusivity. In CLM-KPP, the eddy diffusivity is estimated separately for the lake boundary layer and lake interior. In the lake boundary layer, the eddy diffusivity is not determined by local gradient of mean variables, but it is determined by surface forcing and the boundary layer depth. The non-local effect is taken into account by estimating the

boundary layer depth first, and the eddy diffusivity is specified with a prescribed profile in the boundary layer. In the lake interior, the mixing is generally weak and associated with internal wave activity and shear instability. From our point of view, the major shortcomings of CLM-ORG are that it does not consider a boundary layer for eddy development, and it requires an ad hoc parameter to enhance the estimated eddy diffusivity. In the KPP scheme, an explicit inclusion of an ad hoc enlarging parameter is avoided. The KPP scheme was tested for different time scales, diurnal change, seasonal cycle, and single event for different locations (Large et al. 1994). We have also conducted more simulations for other lakes with quite different environment settings, e.g. Nam Co at Tibetan Plateau with a focus on its long term change, and the results are presented below.

For Nam Co, located in the Tibetan Plateau, we conducted simulations at a 10-km spatial resolution over the period of 2003 through 2012. Our simulations showed that the lake WST simulations with CLM-KPP were significantly improved when compared with CLM-ORG simulations. We have added simulations and analysis for Nam Co to the manuscript:

"We validated both CLM-ORG and CLM-KPP with the monthly Moderate Resolution Imaging Spectroradiometer (MODIS) data for Nam Co by conducting 10-km spatial resolution simulations for this lake over the period of 2003 through 2012. We can see that CLM-KPP improved WST simulations averaged over the entire lake (34 model grid cells) when compared with the CLM-ORG simulations (Fig. R1). The RMSE of WST decreased from 4.58 ˚C with CLM-ORG to 2.23 ˚C with CLM-KPP, and the R increased from 0.90 to 0.96 at the same time.

The differences in the mixing coefficients of CLM-KPP and CLM-ORG cause the difference in WST simulations. We averaged the $K_w^{ORG}$ and $K_w^{KPP}$ over the water columns with the depth greater than 25 m for Nam Co (Fig. R2), and the total of such columns were 28 out of 34 for this lake. Figure R2 indicated that $K_w^{KPP}$ was slightly smaller than $K_w^{ORG}$ mostly in the mixed layer of the lake during summer time. In the deeper part of the lake, $K_w^{KPP}$ was much smaller than $K_w^{ORG}$ during summer time. In the spring and fall seasons, $K_w^{KPP}$ was significantly larger than $K_w^{ORG}$ where the buoyancy flux may contribute strongly to $K_w^{KPP}$. During the winter time when the lake froze, both CLM-KPP and CLM-ORG were set to use $K_w^{ORG}$. We can see that the most significant improvements in WST for Nam Co occurred during the ice-free seasons when the KPP was activated."

[Figure]

Figure R1. Time series over the period of 2003 through 2012 of monthly WST observations from MODIS (black starred line) and simulations with CLM-ORG (blue line) and CLM-KPP (red line) (Unit: ˚C).

[Figure]

Figure R2. Simulated (a) $log_{10} K_w^{ORG}$ with CLM-ORG, (b) $log_{10} K_w^{KPP}$ with CLM-KPP (Unit: m$^2$/s) averaged over water columns with depth greater than 25 m (28 of 34 grid cells), and (c) differences between $log_{10} K_w^{KPP}$ and $log_{10} K_w^{ORG}$ ($log_{10} K_w^{KPP} - log_{10} K_w^{ORG}$).

In CLM-KPP, the eddy diffusivity formulation is different for the boundary layer and lake interior. In the lake boundary layer, the eddy diffusivity is related with boundary layer depth and surface forcing. In the lake interior, the eddy diffusivity is relatively weak, associated with internal wave activity and shear instability. Overall the CLM-KPP can enhance the eddy diffusivity during spring and fall and maintain weak eddy diffusivity in the lake interior during summer when stratification is strong. The outcome of the CLM-KPP eddy diffusivity is an improved WST simulation.

For Fog3 Lake in Alaska, numerical experiments were conducted for CLM-ORG with enhanced wind. Figures R3 and R4 showed simply providing larger winds could not significantly improve CLM-ORG simulations for this lake (Table R1).

When stronger wind is used, the CLM-ORG can simulate the mixing event around 16 Aug. However, the strong wind causes WST to have a negative bias, presumably caused by heat loss from the lake. Thus, as shown in the manuscript, the CLM-KPP provides a better parameterization of eddy diffusivity and improved lake temperature simulations.

[Figure]

Figure R3. WST observations (black line) and CLM-ORG simulations with the default wind data (red line), with wind data 2-fold increased (blue line), with wind data 1.5-fold increased (green line), and with wind data 1.8-fold increased (grey line) (unit: ˚C).

[Figure]

Figure R4. Lake temperature profiles of (a) observations and CLM-ORG simulations with (b) the default wind data, (c) with wind data 2-fold increased, (d) with wind data 1.5-fold increased (green line), and (e) with wind data 1.8-fold increased (grey line) (unit: ˚C).

Table R1. RMSE (˚C) and R of the temperature profile simulations with CLM-ORG, the case with wind 2-fold increased, the case with wind 1.5-fold increased, and the case with wind 1.8-fold increased for Fog3 Lake for the periods of 1 July–15 August and 16–31 August 2018.

|  | 1 July–15 August, 2018 | | 16–31 August, 2018 | |
| --- | --- | --- | --- | --- |
|  | RMSE (˚C) | R | RMSE (˚C) | R |
| CLM-ORG | 1.1 | 0.93 | 1.4 | 0.57 |
| wind×2 | 2.0 | 0.83 | 0.5 | 0.63 |
| wind×1.5 | 0.83 | 0.97 | 1.1 | 0.59 |
| wind×1.8 | 2.0 | 0.84 | 0.6 | 0.62 |

We also mentioned the convective adjustment scheme in the manuscript. The convection scheme works when there exists density instability (Hostetler and Bartlein, 1990).

**Specific comments**

**(1) Lines 88-90: "Researchers have attempted to advance this lake model to more closely reflect reality over the last two decades (Fang and Stefan, 1996; Henderson-Sellers, 1985; Hostetler and Bartlein, 1990; Subin et al., 2012)." Three of four papers cited here do not deal with CLM model.**

**Response:** We deleted this sentence.

**(2) Lines 92-93: "... is the enhanced eddy diffusivity for unresolved mixing processes". All mixing processes in 1D model are unresolved and are parameterized, because only 3D model of sufficiently high resolution simulates turbulence explicitly.**

**Response:** We agreed with this reviewer on this comment. We changed "for unresolved mixing processes" to "to strengthen mixing processes" (Page 3 Line 15)

**(3) Line 98: "0.0012$u_2$" I guess, you can write drag coefficient $C_d$ instead of 0.0012, to make the physical sense of this equality clear.**

**Response:** Yes, we changed "0.0012" to "$C_d$" in the manuscript (Page 3 Line 22 and Eq. (3)).

**(4) Eq. (5): please separate this fraction into two.**

**Response:** Yes, we separated the fraction into two parts (Page 4 Eq. (6)).

**(5) Section 2.1.1: you didn't mention convective adjustment scheme in CLM lake model. It should work during nights**

**in your simulation.**

**Response:** Yes, based on the general comment, we included convective adjustment scheme to the manuscript (Page 3 Line 11 to Line 12).

**(6) Section 2.2: too concise description of the lake. Put more info on climate and landscape conditions, hydrological regime, previous research of the lake.**

**Response:** Yes, we added more description of Fog3 Lake to the manuscript:

Change "Fog 3 Lake, is in Arctic Alaska at (68.67˚ N, 149.10˚ W) (Fig. 1a). In 2018 it had a surface area of 35,230 m$^2$ and a maximum depth of 19.74 m. The lake has a long ice duration, and ice-off is usually in late June, while ice-on typically occurs in early October (Arp et al., 2015)."

to

"Fog3 Lake is in Arctic Alaska at (68.67˚ N, 149.10˚ W) (Fig. 1a). In 2018 it had a surface area of 38,863 m$^2$ and a maximum depth of 21 m. The lake has a long ice duration, and ice-off is usually in late June, while ice-on typically occurs in early October (Arp et al., 2015). Around this lake, the mean annual air temperature is about ~ –6 ℃, and the mean annual precipitation is ~ 200 mm (Ping et al., 1998). This kettle lake is surrounded by lower hills covered mainly with shrubs and tundra. Due to the treeless landscape, there are no shielding effects on the wind. In addition, Fog3 Lake is formed by glaciers, and has less connection to other surrounding surface waters." (Page 6 Line 6 to Line 12).

**(7) Line 154: "wind-only driven scheme". Again (see above), it is incorrect to state that basic CLM lake model includes only wind forcing, as it accounts for both stable and unstable stratification.**

**Response:** Yes, see the response for the general comment.

**(8) Section 2.3: I would add more info on the organization of measurements. Is there a mast on a lake? Which organization runs measurements? Any relevant references?**

**Response:** Fog3 Lake is about 1.5 km from Toolik Field Station (68°37.796' N, 149°35.834' W), in the northern foothills of the Brooks Mountain Range, Alaska (https://toolik.alaska.edu/edc/abiotic_monitoring/index.php) (Page 6 Line 18 to Line 20).

**(9) Line 173: "estimates a stratified lake": sounds badly, please rephrase.**

**Response:** Yes, we changed this sentence to **"CLM-KPP accurately captured the mixing event (Fig. 3c), while CLM-ORG produced strong stratification in the upper part of the lake throughout the simulation period (Fig. 3b)"** (Page 7 Line 28 to Line 29).

**(10) Table 1 is too small, you can easily present those numbers directly in text.**

**Response:** We separated our entire simulation period for Fog3 Lake into the before and after mixing periods and calculated RMSE and R for these two periods (Table R2; Table 1 in the manuscript). We can see that CLM-KPP remarkably improved the water mixing simulations in Fog3 Lake when compared with CLM-ORG.

Table R2. RMSE (˚C) and R of the temperature profile simulations with CLM-ORG and CLM-KPP for Fog3 Lake for the periods of 1 July–15 August and 16–31 August 2018.

|  | 1 July–15 August, 2018 | | 16–31 August, 2018 | |
| --- | --- | --- | --- | --- |
|  | RMSE (˚C) | R | RMSE (˚C) | R |
| CLM-ORG | 1.1 | 0.93 | 1.4 | 0.57 |
| CLM-KPP | 1.3 | 0.92 | 0.3 | 0.99 |

**(11) Lines 183-184: "Thermal forcing played a vital role in this enlarged diffusivity, which was considered only in CLM-KPP and not in CLM-ORG." See my comment 7 above and general comment.**

**Response:** Yes, see the response for the general comment.

**(12) Line 188: "10$^{-7}$" please put units and elsewhere in the document.**

**Response:** Yes, we put units and elsewhere in the manuscript (Page 8).

**(13) Line 188: "was the product" It is not product, but a sum.**

**Response:** Yes, we changed "product" to "sum" in the manuscript (Page 8 Line 11).

**(14) Lines 198-201: two sentences, stating almost the same.**

**Response:** The first sentence states $N^2$, while the second sentence states the water stratification (Page 8 Line 18 to Line 21).

**(15) Line 238: "absorbed solar radiation". It is radiation flux.**

**Response:** Yes, we modified "absorbed solar radiation" to "radiation flux" in the manuscript (Page 11 Line 3).

**(16) Lines 239-240: "total eddy diffusivity". Better: total diffusivity.**

**Response:** Yes, we modified "total eddy diffusivity" to "total diffusivity" in the manuscript (Page 11 Line 4).

**(17) Eq. (A3): a0, a1, ... Better to put numbers into subscript ($a_0$, $a_1$, ...).**

**Response:** Yes, we put numbers into subscript accordingly in the manuscript (Page 11 Line 7).

**(18) Eq. (A4) (both equations): there is a derivative sign in numerator and not in denominator.**

**Response:** Yes, we made it more clearly in the manuscript (Page 11 Line 8 to Line 10).

**(19) Line 244: Not clear, what is $\vartheta(h)$? You say, it is "water diffusivity". But, water diffusivity is $K_w$. There are also molecular diffusivity, background diffusivity, diffusivity caused by internal waves ... all denoted differently above.**

**Response:** $\vartheta(h)$ refers to the total diffusivity of water, a sum of molecular diffusivity, background diffusivity, diffusivity caused by internal waves. We made it more clearly in the manuscript (Page 11 Line 8).

**(20) Line 246: replace "buoyancy difference" by "buoyancy".**

**Response:** Yes, we replaced "buoyancy difference" by "buoyancy" in the manuscript (Page 11 Line 11).

We thank the reviewer 2 very much for the valuable comments on our manuscript. The comments (bolded) are addressed below.

**First, thank you for sharing your work. This is a very interesting study! You present a method of improving the thermal mixing of lakes in the Community Land Model (CLM). The new method introduced into CLM is K profile parameterization (CLM-KPP), a method utilized in ocean modeling. The current CLM vertical mixing scheme (CLMORG) assumes wind is the primary forcing in thermal mixing of lakes. KPP uses wind and surface thermal forcing to simulate lake temperatures. The model did not improve until a mixing event occurred on 16-31 August. CLM-ORG predicted a continued stratification of lake temperature from 16-31 August. CLM-KPP correctly estimated when and the magnitude at which the thermal mixing event would occur from 16-31 August. You provide a thorough analysis as to how thermal forcing within CLM-KPP was able to correctly predict that the mixing would occur. However, I believe there a couple of points that would enhance this work.**

**Major Comments 1. The study seems limited using only one lake and a very narrow time frame. I would recommend the inclusion of several study locations and/or a longer period of analysis to get a better sense of the implications of using CLM-KPP over CLM-ORG. Right now the impact of the study feels limited given that only one location is examined for a two month period during the same season.**

**Response:** We chose another lake, Nam Co, to evaluate CLM-ORG and CLM-KPP. We validated both CLM-ORG and CLM-KPP with the monthly Moderate Resolution Imaging Spectroradiometer (MODIS) data for Nam Co by conducting 10-km spatial resolution simulations for this lake over the period of 2003 through 2012. We can see that CLM-KPP improved WST simulations averaged over the entire lake (34 model grid cells) when compared with the CLM-ORG simulations (Figs. R1 and R2). The RMSE of WST decreased from 4.58 ˚C with CLM-ORG to 2.23 ˚C with CLM-KPP, and the R increased from 0.90 to 0.96 at the same time. We have added simulations and analysis for Nam Co to the manuscript.

**2. Related to 1, you do not provide an analysis of how the stratification beginning on 16 Aug better informs ecosystem, meteorological, or climatological analysis for the lake. A better discussion of implications of capturing this mixing, particularly if any were observed, would enhance this work.**

**Response:** Stratification plays an important role in lake production and food webs. Stratification and warmer epilimnion temperatures create conditions necessary for phytoplankton production. Also, when Arctic lakes become strongly stratified, the hypolimnion can become anoxic, which in turn increases nutrient recycling and leads to elevated production the following spring (O'Brien et al., 2005). Increased food availability and warmer lake temperatures in the epilimnion from stratification increase arctic char growth. Finally, simulations of stratification date and epilimnion temperature are used in

bioenergetic models to estimate fish growth and consumption and better understand Arctic char production with global environmental change (Budy and Luecke, 2014) (Page 2 Line 6 to Line 12).

**3. Line 169-180: You discuss how RMSE and correlation (R) improved with CLM-KPP only slightly for the entire simulation period. I suggest that since you use these metrics, divide the calculation of these metrics into a before and after the mixing event occurs. This would strengthen your point. You should then note this in the abstract and conclusions to better illustrate the impact that CLM-KPP has in the simulation.**

**Response:** Based on this comment, we separated our entire simulation period for Fog3 Lake into the before and after mixing periods and calculated RMSE and R for these two periods (Table R2; Table 1 in the manuscript). We can see that CLM-KPP remarkably improved the water mixing simulations in Fog3 Lake when compared with CLM-ORG.

**Minor comments**

**Line 100: Please define phi.**

**Response:** Actually, we define phi (φ) in Page 3 Line 22 "$k^*$ is related to latitude φ" in the manuscript.

**Line 161: How did you decided upon the 24 layers you specify?**

**Response:** The depth for this lake was set at 20 m in both models. Observed lake temperatures for Fog3 Lake are for lake depths of 0, 1, 2, 3, 4, 5, 6, 7, 8, 10, 12, 14, and 16 m. The lake model has 10 lake layers by default, and the center point depths of these layers are 0.05, 0.3, 0.9, 1.9, 3.3, 5.1, 7.5, 10.3, 13.79, and 17.94 m generated automatically by the layering scheme in the model based on the input lake depth. For this study, we tried to keep each layer thin in the top part of the lake to reflect diurnal cycles (layers 1–5) in both CLM-ORG and CLM-KPP. Below layer 5, we used mostly the observed points to layer the rest of the lake column. Finally, we produced 24 layers for the entire lake column in both models, and the center point depths of these lake layers are 0.05, 0.15, 0.25, 0.35, 0.45, 1, 2, 3, 4, 5, 6, 7, 8, 9, 10, 11, 12, 13, 14, 15, 16, 17, 18, and 19.25 m, respectively (Page 7 Line 8 to Line 15). As shown in the Fig. R5 and Table R3, the simulations of CLM-ORG with both 10 and 24 layers were very similar, while the simulations of CLM-KPP with 24 layers were closer to observations than those with 10 layers when the water mixing event occurred.

[Figure]

Figure R5. Lake temperature profiles of (a) observations, simulations of (b) CLM-ORG and (c) CLM-KPP with 24 layers and simulations of (d) CLM-ORG and (e) CLM-KPP with 10 layers (unit: ˚C).

Table R3. RMSE (˚C) and R of the temperature profile simulations of CLM-ORG and CLM-KPP with 10 and 24 layers for Fog3 Lake over the periods of 1 July–15 August and 16–31 August 2018.

|  |  | 1 July–15 August, 2018 | | 16–31 August, 2018 | |
| --- | --- | --- | --- | --- | --- |
|  |  | RMSE (˚C) | R | RMSE (˚C) | R |
| 10 model layers | CLM-ORG | 1.0 | 0.94 | 1.4 | 0.58 |
|  | CLM-KPP | 1.2 | 0.94 | 0.7 | 0.90 |
| 24 model layers | CLM-ORG | 1.1 | 0.93 | 1.4 | 0.57 |
|  | CLM-KPP | 1.3 | 0.92 | 0.3 | 0.99 |

**Responses to Dr. Zeli Tan:**

We thank Dr. Zeli Tan very much for the constructive and helpful comments on our manuscript. The comments (bolded) from the reviewer Dr. Zeli Tan are fully addressed in the following.

**It is an interesting study. Because a 1-D lake model is still much needed to understand the impact of climate changes on global lake systems, a parameterization method that could improve the simulation of lake mixing process will be much valued. But I suggest that the manuscript can be improved in the following directions. First of all, the comparison between CLM-ORG and CLM-KPP is not exhausted, to day the least. In Subin's CLM-ORG paper, he actually tested the model over a pair of lakes around the globe. In fact, the CLM-ORG performance on high-latitude lakes which this study focused on was not the worst. Thus, the method can become much more valuable if the authors can apply this method to some more lakes, especially those deep and large lakes.**

**Response:** Thanks for the comments. We chose another lake, Nam Co, to evaluate CLM-ORG and CLM-KPP. We validated both CLM-ORG and CLM-KPP with the monthly Moderate Resolution Imaging Spectroradiometer (MODIS) data for Nam Co by conducting 10-km spatial resolution simulations for this lake over the period of 2003 through 2012. We can see that CLM-KPP improved WST simulations averaged over the entire lake (34 model grid cells) when compared with the CLM-ORG simulations (Figs. R1 and R2). The RMSE of WST decreased from 4.58 ˚C with CLM-ORG to 2.23 ˚C with CLM-KPP, and the R increased from 0.90 to 0.96 at the same time. We have added simulations and analysis for Nam Co to the manuscript.

**Second, more information about the study lake is needed. Is Fog3 Lake a glacial lake or a thermokarst lake? How was the surface friction velocity derived for this lake? Are the effects of lake fetch and wind shielding considered? What is the lake's light attenuation coefficient?**

**Response:** Fog3 Lake is a glacial lake. In CLM-ORG, the surface friction velocity $w^*$ (m/s) is calculated as:

$$w^* = 0.0012u_2 \tag{R1}$$

where $u_2$ is the 2-m wind speed (m/s).

While in CLM-KPP, the surface friction velocity $u^*$ (m/s) is calculated as (Large and Pond, 1982):

$$u^{*2} = \frac{\rho_a}{\rho} C_D U^2 \tag{R2a}$$

$$10^3 C_D = \frac{2.70}{U} + 0.142 + 0.0764U \tag{R2b}$$

where $\rho_a$ and $\rho$ are the air and lake water densities (kg/m$^3$) respectively, $C_D$ is the drag coefficient and $U$ is the 10-m wind speed (m/s). The effect of the lake fetch was considered in our simulations. In the CLM-ORG, the lake fetch F (m) (Hutchinson, 1957; Wetzel and Likens, 1991) is:

$$F = \begin{Bmatrix} 100, & D < 4 \\ 25D, & D \geq 4 \end{Bmatrix} \qquad (R3)$$

where D is the water depth. We also used this function in CLM-KPP.

In this study, wind shielding was not considered. Actually, the Toolik meteorological station providing the wind data is ~1.5 km away from Fog3 Lake, although there are no buildings or trees between the Toolik station and the lake. Thus, the wind shielding effects are not significant. The light extinction coefficient η (m$^{-1}$) is a function of depth (m) (Hakanson, 1995):

$$η = 1.1925D^{-0.424} \qquad (R4)$$

In this study, with the lake depth (D) of 20 m for Fog3 Lake, η is about 0.33 m$^{-1}$.

**Third, how are CLM-ORG and CLM-KPP calibrated in this study? I know that CLM-ORG has a water mixing parameter that can be used to increase diffusivity for those deep lakes. Can the parameter values of CLM-KPP described here be applied to other lakes?**

**Response:** Both CLM-ORG and CLM-KPP were not calibrated in this study. Yes, the water mixing parameter in CLM-ORG can be increased to generate stronger water mixing for deep lakes (Gu et al., 2013). Here, we increased the water diffusivity (Eq. (1) in the manuscript) by 10 and 100 times in CLM-ORG and conducted additional simulations for Fog3 Lake as shown in Figs. R6 and R7. We can see that CLM-ORG was still unable to reproduce the observed lake temperatures with the enlarged water diffusivity. Again, we did not adjust any parameters in CLM-KPP when we performed simulations for Fog3 Lake, and the same parameters were applied to the simulations for Nam Co. We see that CLM-KPP more realistically captured the water mixing in Nam Co than CLM-ORG (Figs. R1 and R2).

[Figure]

Figure R6. Lake WST observations (black line), simulations with CLM-ORG (red line), and simulations with $K_w^{ORG}$ multiplied by 10 (blue line) and 100 (green line), respectively.

[Figure]

Figure R7. Lake temperature profiles of (a) observations, (b) simulations with CLM-ORG, and simulations with $K_w^{ORG}$ multiplied by (c) 10 and (d) 100.

**Forth, I am surprised that the case study did not cover the period of spring water mixing which can have large biogeochemical impacts for high-latitude lakes.**

**Response:** Lake temperature data and some of the atmospheric forcing data for Fog3 Lake are available only for July and August 2018. However, our additional simulations with CLM-ORG and CLM-KPP for Nam Co covered the period of 2003-2012, which included the spring season (Figs. R1 and R2). Our simulations with CLM-KPP were closer to observations than those with CLM-ORG for almost the entire simulation period including the spring seasons.

**Thanks for the authors to address my comments patiently. Overall, the response is great.**

**Just to remind that MODIS data is probably not good for lake model validation at specific lakes, especially at the spring and fall mixing periods when the rapid change of weather would introduce significant uncertainties (such as cloud cover). Thus, the uncertainty of MODIS data need to be acknowledged.**

**Responses:** Yes, we acknowledged the uncertainties of MODIS data in the manuscript (Page 7 Line 2 to Line 5). Previous studies have verified MODIS WST data for lakes with *in situ* observations (Crosman and Horel, 2009; Schneider et al., 2009). Zhang et al. (2014) found that the nighttime WST of MODIS for Nam Co had a 0.89 correlation coefficient and a − 1.4 ℃ bias when compared with surface observations. All these studies show that the MODIS WST has acceptable accuracy for studying lake thermal processes.

**In addition, I do not think that the overestimation of surface temperature by CLM-ORG in summer is due to lack of mixing (Fig. R1). The other causes, such as the representation of latent and sensible heat, need to be acknowledged.**

**Responses:** Yes, Figure R2 showed that $K_w^{KPP}$ was slightly smaller than $K_w^{ORG}$ mostly in the mixed layer of the lake during summer time. In the deeper part of the lake, $K_w^{KPP}$ was much smaller than $K_w^{ORG}$ during summer time. In the spring and fall

seasons, $K_w^{KPP}$ was significantly larger than $K_w^{ORG}$.

In CLM-KPP, the eddy diffusivity formulation is different for the boundary layer and lake interior. In the lake boundary layer, the eddy diffusivity is related with boundary layer depth and surface forcing. In the lake interior, water diffusivity is relatively weak, associated with internal wave activity and shear instability. Overall, CLM-KPP enhances the water diffusivity during the spring and fall and maintains weak water diffusivity in the lake interior during the summer when stratification is strong when compared to CLM-ORG. In addition, the overestimated WST with CLM-ORG before the summer affects the energy budget at the lake surface, which further influences lake temperature simulations during the summer.

[revised manuscript text omitted]